# Temporal novelty detection and multiple timescale integration drive *Drosophila* orientation dynamics in temporally diverse olfactory environments

**Viraaj Jayaram**[1,2,3�], **Aarti Sehdev**[1,2�], **Nirag Kadakia**[1,2,4], **Ethan A. Brown**[1,2,5], **Thierry Emonet**[1,2,3,6]*

**1** Department of Molecular, Cellular, and Developmental Biology, Yale University, New Haven, Connecticut, United States of America, **2** Quantitative Biology Institute, Yale University, New Haven, Connecticut, United States of America, **3** Department of Physics, Yale University, New Haven, Connecticut, United States of America, **4** Swartz Foundation for Theoretical Neuroscience, Yale University, New Haven, Connecticut, United States of America, **5** Yale College, Yale University, New Haven, Connecticut, United States of America, **6** Interdepartmental Neuroscience Program, Yale University, New Haven, Connecticut, United States of America

ží These authors contributed equally to this work.
* thierry.emonet@yale.edu

**Data Availability Statement:** All fly trajectory information and software used for data analysis and figure generation in this study are available on

## Abstract

To survive, insects must effectively navigate odor plumes to their source. In natural plumes, turbulent winds break up smooth odor regions into disconnected patches, so navigators encounter brief bursts of odor interrupted by bouts of clean air. The timing of these encounters plays a critical role in navigation, determining the direction, rate, and magnitude of insects' orientation and speed dynamics. Disambiguating the specific role of odor timing from other cues, such as spatial structure, is challenging due to natural correlations between plumes' temporal and spatial features. Here, we use optogenetics to isolate temporal features of odor signals, examining how the frequency and duration of odor encounters shape the navigational decisions of freely-walking *Drosophila*. We find that fly angular velocity depends on signal frequency and intermittency–the fraction of time signal can be detected– but not directly on durations. Rather than switching strategies when signal statistics change, flies smoothly transition between signal regimes, by combining an odor offset response with a frequency-dependent novelty-like response. In the latter, flies are more likely to turn in response to each odor hit only when the hits are sparse. Finally, the upwind bias of individual turns relies on a filtering scheme with two distinct timescales, allowing rapid and sustained responses in a variety of signal statistics. A quantitative model incorporating these ingredients recapitulates fly orientation dynamics across a wide range of environments and shows that temporal novelty detection, when combined with odor motion detection, enhances odor plume navigation.

Dryad at Jayaram, Viraaj et al. (2023), Data and software for "Temporal novelty detection and multiple timescale integration drive Drosophila orientation dynamics in temporally diverse olfactory environments", Dryad, Dataset, https://doi.org/10.5061/dryad.pzgmsbcrg. Fly lines used in experiments are available upon request to Thierry.emonet@yale.edu as long as they are maintain in our lab.

**Funding:** NK was supported by a postdoctoral fellowship through the Swartz Foundation for Theoretical Neuroscience, by postdoctoral fellowships NIH F32MH118700 and NIH K99DC019397. VJ and TE were partially supported by the Program in Physics, Engineering, and Biology at Yale. The project was supported by TE's setup funds from Yale University. The funders had no role in study design, data collection and analysis, decision to publish or preparation of the manuscript.

**Competing interests:** The authors have declared that no competing interests exist.

## Author summary

Olfactory navigation is essential for insects to find food and mates but challenging because complex wind flows break odor plumes up into discrete and intermittent packets. The timing of encounters with these packets is crucial for navigation, affecting when insects reorient. To decide where to reorient, insects extract directional information from the wind, the odor gradient, and the motion of the odor packets, by comparing signals between their two antennae. Here we ask how the frequency and duration of odor encounters drive reorientation. To isolate the role of odor timing we use a virtual reality setup in which freely walking flies experience a constant wind direction along with uniform flashes of light–virtual odor packets–that activate their odor sensors uniformly, thus removing all odor directional cues. We find that flies are much more likely to respond to an individual odor encounter if the time since the previous encounter is greater than ~2s. We show in simulations how this temporal novelty detection, when combined with odor motion-sensing, enhances navigation. In turbulent plumes odor packets tend to arrive in clumps. Our findings suggest flies can respond differently to the beginning of a clump than to fluctuations within to enhance navigation.

## Introduction

Olfactory navigation is a challenging task, owing to the complexity and variability of natural odor scenes. The distribution of odors in nature depends sensitively on the physical properties of the environment, such as airflow and proximity to surfaces and boundaries, creating a diversity of signals varying in their spatial and temporal statistics [1–3]. Animals such as insects must extract relevant odor information from these complex landscapes, and use it to inform rapid behavioral decisions to progress toward the odor source.

In diffusion-dominating odor environments such as near food-laden surfaces, animals can locate odor sources by sampling concentration gradients temporally [4–7] and spatially [8,9]. In stronger airflows, further away from boundaries, and above rough terrain such as rocks or trees, complex airflows break up smooth odor regions into discrete packets and filaments swept along by the wind [3,10–16]. As a result, animals experience discrete encounters with odor packets separated by blanks (moments when odor concentration is below detection threshold). The durations of these encounters can span a wide range of timescales [11]. Under such conditions, insects navigate by orienting upwind within the odor and moving crosswind or downwind when the odor is lost, in an attempt to regain the plume [17]. Similar behaviors (with some variations between species) are observed in laboratory experiments with walking and flying moths [10,17–23] and fruit flies [24–27]. We recently discovered that flies can also detect the direction of motion of odor signals, by resolving inter-antennal concentration differences over time. Odor motion provides a directional cue complementary to the wind, and is especially useful in turbulent plume navigation [28]. In sum, despite variations between species and locomotive regimes, the general picture of insect odor navigation is that the wind, along with odor gradients and the recently-discovered odor motion, indicate *directions* of interest, while the timing of odor encounters and the odor identity indicate *when* to turn and how much to bias the new orientation in the directions of interest.

Careful analyses of moth turning responses following odor encounters have implicated the frequency of odor encounters as a key driver of upwind progress [21–23]. Frequency-driven turning is also observed in walking flies navigating complex odor plumes when odor encounters are brief (~100 ms) and frequent [27]. Conversely, flies experiencing longer and sparser

odor encounters progress upwind by integrating the odor concentration over time–thus responding to odor intermittency-the fraction of time signal can be detected-or duration [26,29,30], rather than encounter onset time or frequency. Thus, insects are clearly able to sense and process various temporal features of the odor signal during plume navigation; moreover, this broad and versatile sensing capability has been shown theoretically to enable efficient source localization across a diversity of plume structures [31–33]. Still, how these multiple features are precisely weighted within a single navigation strategy, and whether the strategy itself modulates as signal statistics change, remains unclear.

In this study, we address these questions using an optogenetic assay developed in previous studies [27,28]. We present spatially uniform but temporally-structured fictive odor stimuli to freely-walking blind *Drosophila melanogaster* in a steady laminar flow. In addition to decoupling odor signal from wind, the spatially uniform stimulus removes both the effect of behavioral feedback on the received odor signal, and any bilateral differences between antennae in timing or intensity from the odor encounters [8,9,28]. Thus, no directional information from the odor signal (odor gradient and odor motion) is available to the flies, which must navigate using the temporal features of the odor signals and the fixed wind direction alone.

Our main findings are the following. i) Fly angular velocity is controlled by the frequency and intermittency of odor encounters, but not their duration. ii) Flies demonstrated a "temporal novelty detection" in turn rate and turn speed: they responded more strongly to signal onset when the prior period without stimulus was longer than ~2 seconds. As in previous studies [26] we also observed an "offset response" in turning behavior, which peaks both at the end of a long odor encounter or a block of many encounters at high intermittency. Importantly these two features combine to smoothly transition the behavioral response of the flies between low and high frequency regimes. iii) The upwind bias of turns (likelihood to orient upwind when turning) was independently modulated by frequency and intermittency of the signal. This dependency resulted from a rapid increase in upwind bias at the onset of odor pulses, followed by a slower decay at the offset, and allowed for strong upwind responses across a wide range of temporally diverse odor environments. We incorporated these findings into a model combining temporal novelty and offset responses together with a two-timescale integrator. This versatile but parsimonious model could recapitulate turn rate, turn speed, and upwind bias across the full spectrum of temporally diverse environments, thus unifying results from previous studies into one framework [26,27,31]. Finally, we show in agent-based simulations how the temporal novelty detection response can be combined with odor motion sensing to improve navigation performance in more complex odor environments.

## Results

### An optogenetic setup to examine the olfactory response of free-walking flies to the temporal features of odor signals

To investigate how fly navigation decisions depend directly on the temporal features of odor signals, we created an optogenetic stimulus (henceforth referred to as a 'fictive' plume) that had only a temporal component yet drove clear navigational responses. Using the wind tunnel walking assay previously described [27] (**Fig 1A**), we presented a temporally variable but spatially uniform optogenetic odor stimulus (**Fig 1B**) to freely-walking blind flies that expressed *Chrimson* in their olfactory receptor neurons (ORNs) (*w;+;Orco-GAL4, w;gmr-hid;UAS-20XChrimson*), from here referred to as *Orco>Chr* mutants. The stimulus was presented in a 15s ON block, where the entire arena was illuminated with a uniform red light stimulus (same intensity as in [28]) and flashed regularly at a frequency of 2 Hz and duration of 0.05s

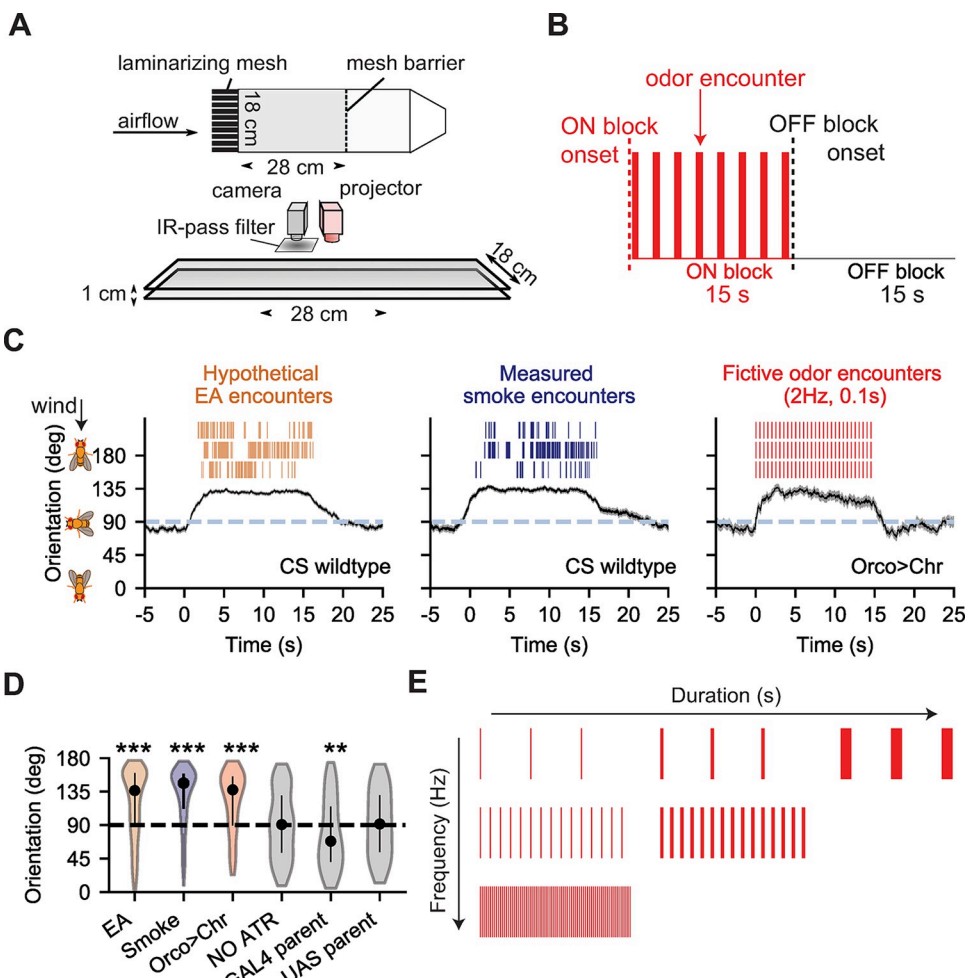

**Fig 1. Optogenetic stimulation with high frequency fictive odor pulses drives similar navigation behaviors as real odor plumes. A.** Top and side view of the fly walking assay. Red light (same intensity as in [28]) is projected from above as fictive odor stimuli, uniformly illuminating the entire arena with high temporal (< 6 ms) precision. Wind speed is 150 mm/s like in [27]. **B.** Four sequential ON and OFF blocks are presented to flies (15 s per block). During the ON block, flashes of red light are presented with a given duration and frequency. No flashes are presented during the OFF block. **C.** Orientation of flies during the ON and OFF block, for odor plumes of ethyl acetate (EA; left column, orange) and smoke (middle column, blue) with turbulent wind (data from [27]) and fictive odor plume presentation at 2 Hz, 0.1 s (right column, red) with laminar wind. Black: mean response over all trajectories. Grey shade indicates SEM. Only walking flies (ground speed > 2 mm/s) were included. Canton-S flies were used for both smoke (114–282 trajectories per frame) and EA (243–582 trajectories per frame) plume presentation. *w;gmr-hid/+;Orco-GAL4/UAS-20XChrimson* mutants (60–99 trajectories per frame) were used for fictive odor plume presentation. We reflected orientations greater than 180˚ so that orientations were always between 0˚ (directly downwind) and 180˚ (directly upwind). Thus, a uniform spread of orientations results in an average orientation of 90˚. **D.** Mean orientation response across all trajectories during 3–12 s of ON block. EA: Canton-S flies in turbulent ethyl acetate plume; Smoke: Canton-S flies in turbulent smoke plume. Orco>Chr: *w; gmr-hid/+; Orco-GAL4/UAS-20XChrimson* in fictive plume. NO ATR: *Orco>Chr* mutant without feeding all-trans retinal, in fictive plume. GAL4 parent: *w;+;Orco-Gal4*, in fictive plume. UAS parent: *w;gmr-hid;UAS-Chrimson*, in fictive plume. Ns are 918, 412, 144, 272, 126 and 114 trajectories respectively. **E.** Full range of optogenetic stimuli presented, varying in encounter frequency (0.2, 0.5, 1.0, 1.5, 1.75, 2, 2.5, 3, 4, 5 Hz) and encounter duration (0.02, 0.05, 0.10, 0.25, 0.50, 1.00 s). Combinations of duration and frequency that created indeterminable flashes (i.e. intermittency > = 1) were excluded.

(consistent with naturalistic complex plumes [27]), followed by a 15s OFF block with no stimulus. A steady unidirectional laminar wind was used as a directional cue for flies to follow. Wind speed was 150 mm/s, matching the wind speed in [27].

In our previous study–which used an identical behavioral setup and genotype–we showed that optogenetically-active flies navigated straight ribbons and complex plumes similarly to real odors [28]. Moreover, light-driven ORN firing responses were well-maintained within their expected physiological range [28]. Many other studies have also successfully used optogenetic stimulation to evoke navigation behaviors (e.g. [34–36]). Here, to further confirm that our optogenetic stimulus drove responses similar to wildtype flies encountering a real odor plume, we examined fly orientation during stimulus presentation. We compared to previous studies in which wildtype *Canton-S* (CS) flies navigated two real odors: ethyl acetate (EA) and smoke [27]. Indeed, flies responding to the optogenetic stimulus showed qualitatively similar navigational trends as flies experiencing real odor plumes, directing their orientation upwind (i.e. towards the fictive odor source) to a very similar degree (**Fig 1C**).

To confirm that *Orco>Chr* mutants were orienting upwind due to the fictive plume stimuli and not some other confounding factor, such as ambient lighting in the experimental arena, we obtained the mean orientation of all trajectories during the ON block (3–12 s) for each environment (CS in EA, CS in smoke, *Orco>Chr* in fictive plume), along with the parental controls of the optogenetically active line (*w;+;Orco-GAL4*, *w;gmr-hid;20XUAS-Chrimson*). *Orco>Chr* mutant responses were additionally measured in the absence of all-trans-retinal (ATR). We compared the mean orientation in these environments to the mean of uniformly distributed headings (which due to the way we reflect orientations results in a mean of 90˚-see **Fig 1** caption) in both laminar and turbulent wind environments (**S1 Fig**). During the ON block, both CS flies in EA or smoke and *Orco>Chr* mutants in the fictive plume oriented more upwind than crosswind (one-sample t-test, EA: 115.7˚±1.4˚, pval < 1e⁻⁶, Smoke: 130.6˚±1.7˚, pval < 1e⁻⁶ *Orco>Chr*: 119.9˚±2.9˚, pval < 1e⁻⁶) (**Fig 1D**). In comparison, the orientation of both the UAS-parental control line and the *Orco>Chr* mutant line without ATR did not differ from uniform orientation (*UAS*: 92.4˚±3.5˚, pval = 0.491, NO ATR: 92.2˚±2.4˚, pval = 0.343), (**Figs 1D, grey and S2**). Interestingly, the GAL4-parental control line oriented more downwind than expected (79.1˚±2.9˚, pval = 0.0002), which we attributed to a mild influence of background visual stimuli, since this parent was not blind.

The similarity in gross behaviors between wildtype flies navigating real odors and optogenetically stimulated flies navigating fictive odors indicates that spatially uniform, dynamic optogenetic stimuli can drive upwind naturalistic plume navigation. This is consistent with previous results that used real odors [26], though an added benefit here is that bilateral information is entirely removed by using full-field optogenetic flashes. Temporal signal variation alone is enough to drive persistent upwind navigation, emphasizing the importance of temporal stimuli features in the absence of spatially-variable concentrations or local gradients.

## Upwind heading correlates with signal frequency and intermittency, but not duration

Next, we asked how frequency, duration and intermittency modulate upwind heading. We generated 45 fictive odor environments with pulse durations between 0.02s and 1s, and pulse frequency between 0.2 Hz and 5 Hz (**Fig 1E**). Intermittency is equal to frequency multiplied by duration, and varied between 0.004 and 0.875 (intermittency is bounded between 0 (signal is never present) and 1 (signal is always present)).

Average upwind heading exhibited common trends across environments (**Fig 2A**). At ON block onset, flies oriented toward the upwind direction (180˚) when the odor was present, and went downwind at odor offset, similar to behavior in real odor plumes (see **Fig 1**). For low frequencies, where there was sufficient time between encounter onsets to distinguish individual encounter responses (i.e. 0.2Hz, 0.5Hz), flies oriented upwind at each encounter onset,

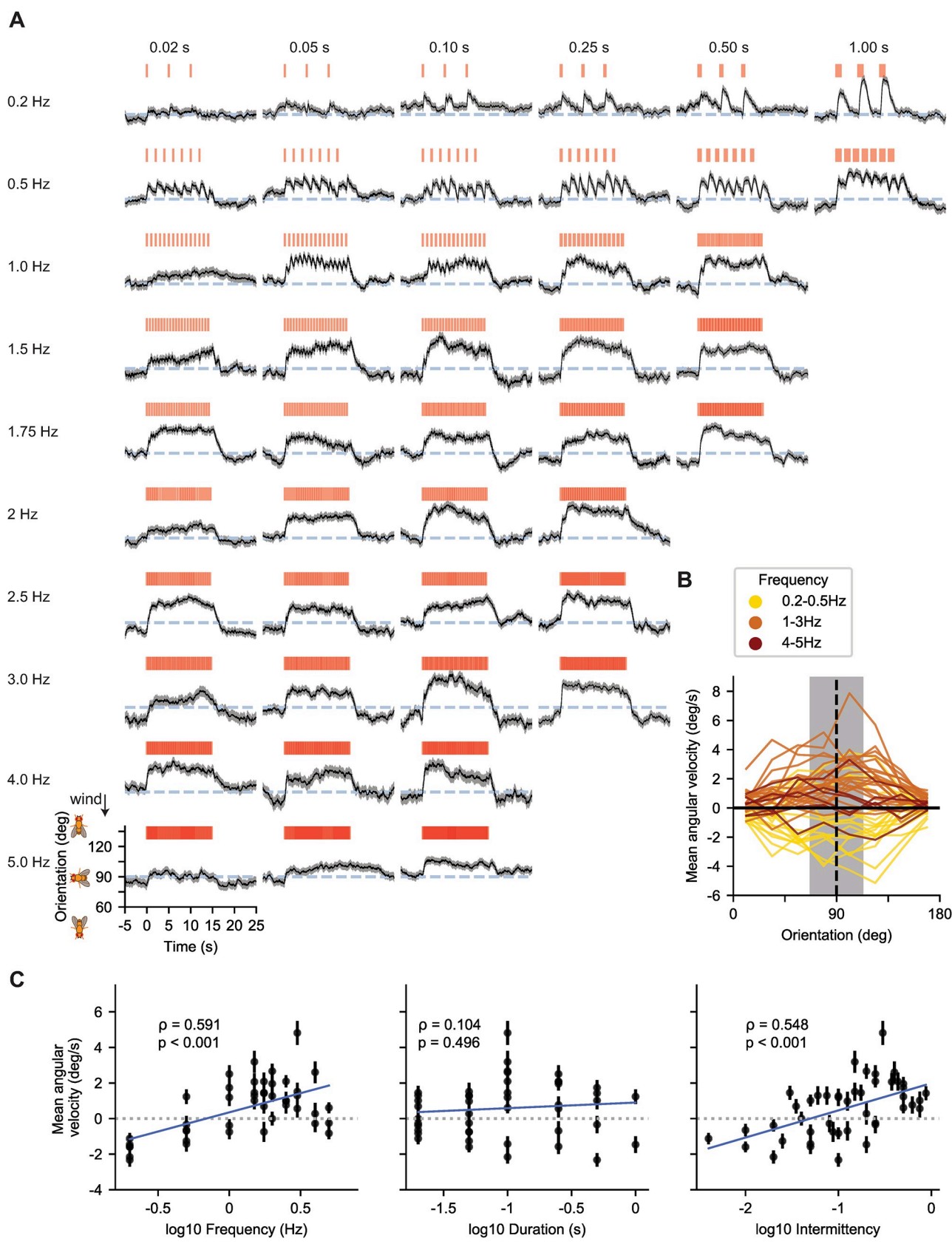

**Fig 2. Flies use the frequency and intermittency of odor signals to navigate upwind across temporally diverse fictive odor environments. A.** Population mean orientation response across 45 fictive odor environments. Each environment projected stimuli with a fixed duration and frequency. Each row represents a different tested frequency: 0.2 Hz, 0.5 Hz, 1 Hz, 1.5 Hz, 1.75 Hz, 2 Hz, 2.5 Hz, 3 Hz, 4 Hz and 5 Hz, from top to bottom. Each column represents a different duration: 0.02 s, 0.05 s, 0.1 s, 0.25 s, 0.5 s and 1 s from left to right. Red bars denote the signal simultaneously encountered by all flies within an experiment (Materials and Methods). Upwind is 180˚, downwind is 0˚. Grey-blue dashed line: crosswind direction (90˚). Black: Population mean orientation; orientation was flipped over 180˚ as before. Grey shading: SEM for each time point (recording rate = 60 Hz). Between 176 and 407 trajectories were recorded per environment. Between 72 and 237 trajectories were recorded per time point across all environments. B. Instantaneous angular velocity of flies as a function of their orientation during the ON block (0–15 s). Upwind is at 180˚, downwind at 0˚. Orientation was split into 8 bins with width 22.5˚. Vertical dashed line indicates crosswind orientation 90˚. Positive (negative) angular velocities correspond to upwind (downwind) turning. Black horizontal line at 0˚/s indicates no change in orientation. Color indicates environment frequency; yellow: low frequencies (0.2 Hz, 0.5 Hz), orange: medium frequencies (1 Hz, 1.5 Hz, 1.75 Hz, 2 Hz, 2.5 Hz, 3 Hz), red: high frequencies (4 Hz, 5 Hz). C. Mean angular velocity of individual flies oriented within the crosswind range (90˚± 22.5˚; grey shading in B) over duration of ON block (0–15 s) as a function of environment frequency (left), duration (middle) and intermittency (right). Each point represents a different environment with defined frequency and duration. Error bars: SEM for that environment. Dotted grey line represents no mean change in orientation. ρ value is the Pearson's correlation coefficient between mean angular velocity and the temporal feature (frequency, duration, intermittency), obtained from linear least-squares regression. Correlations with environment frequency and intermittency were significantly different from 0 (frequency: ρ = 0.63, p < 0.001; intermittency: ρ = 0.55, p < 0.001). Correlation with duration was not significantly different from 0 (ρ = 0.07, p = 0.633).

maintained their upwind orientation for the duration of the encounter, then oriented downwind at the end of the encounter. Beyond 1Hz, individual encounter responses were largely indistinguishable, but we observed that flies drove their orientation upwind for the duration of the ON block, and turned downwind at ON block offset. For low frequencies, average upwind orientation increased with duration, but this effect tapered beyond ~1Hz. Meanwhile, for a given pulse duration, mean orientation increased with frequency up to around 3Hz, before decreasing at very high frequencies. This aligns with previous studies that have shown that olfactory receptor neurons can respond to high frequencies [37]. These trends in mean orientation were observed in both laminar and complex wind conditions (**S3 Fig**).

To quantify how the frequency, duration, and intermittency of odor encounters influence upwind bias, we calculated the instantaneous angular velocity as a function of orientation at each time point during the ON block (Materials and Methods) (**Fig 2B**). Here, angular velocity was signed such that upwind turns were positive and downwind turns were negative. Average angular velocity was nearly zero when flies were oriented upwind or downwind, but became increasingly positive with signal frequency for those oriented crosswind, up to around 3 Hz. Similar trends were found with intermittency, but not duration (**S4 Fig**). Since these trends were most apparent when flies were oriented crosswind (grey region in **Fig 2B**), we pooled the angular velocities over all instances in which flies were oriented within a 45˚ sector around the crosswind (90˚) direction, and calculated correlations with signal frequency, duration, or intermittency (**Fig 2C**). We found a significant positive correlation between angular velocity and either frequency (Pearson's correlation coefficient, R = 0.59, *p<0.001*) or intermittency (R = 0.55, *p<0.001*), but not duration (R = 0.10, *p = 0.496*). This result indicated that flies use odor frequency and intermittency to drive upwind motion, prompting us to examine behavioral models that respond to these particular signal features.

## Turn dynamics exhibit a temporal novelty response and offset response

Fly orientation results from the cumulative effect of individual turns. To understand how temporal features of the odor signal drive turn dynamics, we first explored how the average angular speed (the magnitude of the angular velocity) was modulated during the signal block across environments (**Fig 3A**). For clarity, we focus on four example odor environments chosen from the 45 environments shown in **Fig 2**. We chose these 4 cases to illustrate the different signal and response regimes present in the full dataset (**S5 Fig**).

In low frequency and intermittency environments (e.g. 0.2Hz, 1s), where responses to individual odor encounters could be clearly resolved, the mean angular speed was dynamic and

    

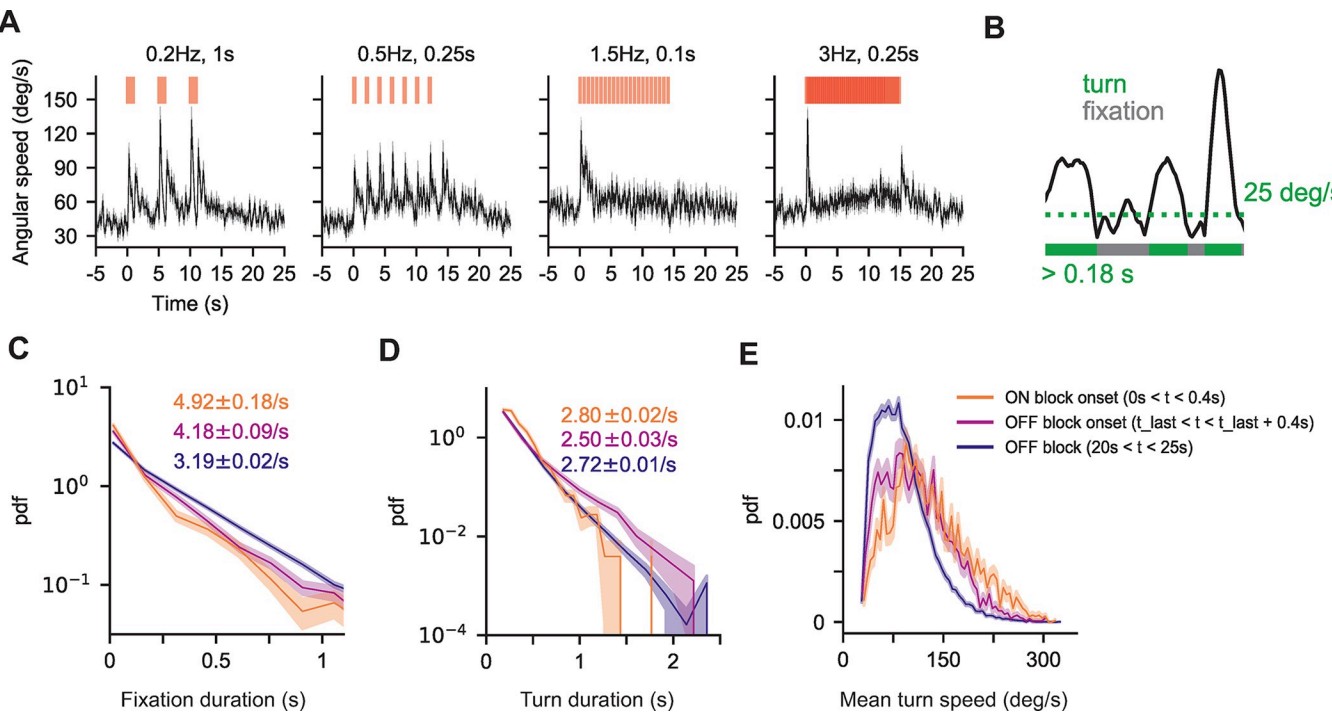

**Fig 3. Flies change orientation in discrete turn events modulated by the signal. A.** Population mean angular speed during stimulus presentation for four conditions– 0.2 Hz, 1 s (left), 0.5 Hz, 0.25 s, (middle-left), 1.5Hz, 0.1s (middle right) and 3 Hz, 0.25 s (right)–obtained from the instantaneous absolute angular velocity of tracked flies (Materials and Methods). Black line: population mean at each time point. Grey shading: SEM at each time point. Red bars: stimulus. 0.2 Hz, 1 s, n = 93–154 trajectories per time point. 0.5Hz, 1s, n = 86–130 trajectories. 1.5Hz, 0.1s, n = 72–120 trajectories. 3 Hz, 0.25 s, n = 105–181 trajectories. **B.** Definition of turn and fixation events. Black: angular speed of an individual fly. Turns must have a minimum angular speed of 25 deg/s (green dotted line) and a minimum duration of 0.18 s (see S7 Fig). **C.** Distribution of fixation durations across all 45 frequency-duration experiments at different times during the experiment. Fixation durations are roughly exponentially distributed with a rate (slope) that depends on signal timing (see S8 Fig). Orange: turn rate during ON block onset (0–0.4 s). Pink: turn rate at OFF block onset (from $t_{last}$ to $t_{last}$+0.5 s) where $t_{last}$ is the time at the end of the last stimulus of a ON block. Purple: baseline turn rate during OFF block (20–25 s). Lines are PDFs. Light shade indicates standard error estimated using bootstrapping (repeats = 1000). **D.** Same as **C** but for the distribution of turn durations. Turn duration distributions do not vary much over time **E.** Distribution of mean angular speed during turns across all 45 frequency-duration experiments at different times. Colors and shading are the same as in **C**. The shifting mean indicates that turns are faster during ON block onset and offset as compared to the baseline turn speed when no stimulus is present.

peaked at each encounter onset (**Figs 3A and S5**). For higher frequency environments >~1.5 Hz, angular speed peaked sharply only at the onset of the ON block, rather than at each individual encounter. It quickly dropped towards the pre-stimulus baseline and remained roughly steady for the remainder of the ON block. We first wondered whether this was because in higher frequency environments, flies had maintained their upwind orientation upon receiving new pulses and hence did not need to reorient upwind. However, we found that in higher frequency environments, even flies facing crosswind or downwind at the onset of later pulses did not show large changes in mean angular speed (**S6 Fig**). This suggests a type of "temporal novelty response", where angular speed will spike at the onset of a new encounter provided the previous odor encounter was in the sufficiently distant past. In high intermittency environments e.g. 3 Hz, 0.25 s, we observed a second large, sharp peak in angular speed at the end of the ON block, which was also seen at the end of individual, long duration encounters (e.g. 0.2Hz, 1s) similar to previous observations [26]. These data suggest that turn dynamics exhibit temporal novelty detection and offset-response, and that they are modulated by both odor frequency and intermittency. While odor offset responses have been seen and quantified before [26], the "temporal novelty response" following an unexpected odor encounter has not yet

been characterized, though it is also observable in previous studies (see Figs 1F and 2A in [26] and Fig 2 and supplement 1B in [27]).

To unravel the decisions underlying these angular speed dynamics, we turned from population averages to individual trajectories. At the level of individual trajectories, changes in angular speed exhibited large, discrete jumps (**Fig 3B**), that occurred both during ON and OFF blocks across all odor environments. Angular speed also underwent small fluctuations that we posited to occur from the fly's walking gait [38] and measurement noise. Following previous work [27,34,39,40], we attributed the large angular changes to turn events, i.e. intentional, large-scale reorientations that align the navigator's heading to the direction of interest. We defined turn events by setting a threshold on angular speed. Events above threshold were called "turns", those below threshold were called "fixations". The threshold (25 deg/s) was chosen to remove small fluctuations that contribute little to the overall change in heading, but keep large angular changes that drive navigation behaviors. We also set a minimum turn duration of 0.18s to remove very short fluctuations in angular speed that were potentially artefacts of the tracking (**S7 Fig** and Materials and Methods).

Having defined turn and fixation events, we examined how the rate, duration, and angular speed of these discrete events, which modulate total angular speed, are influenced by signal statistics. To obtain the turn rate, we note that >95% of fixation events (times between turns) lasted less than 1.5s (**S8 Fig**), and within this range, the distribution of fixation events appeared approximately exponential (**S8 Fig**), suggesting that turn events obeyed a Poisson process. The slope of the distribution, i.e. the turn rate, changed with time (**Fig 3C**). It was high at the onset ($4.92 \pm 0.18$ turns/s) and offset ($4.18 \pm 0.09$ turns/s) of ON blocks, but lower ($3.19 \pm 0.02$ turns/s) during OFF blocks. Turn durations also appeared exponentially distributed, but with a rate that varied less over time (**Fig 3D**). Finally, the mean turn speed exhibited a unimodal distribution that resembled a Gamma distribution with mean that strongly depended on the signal (**Fig 3E**): higher at the onset and offset of the ON block, and lower otherwise. Together, this suggested that angular speed dynamics depended more on changes in turn rate and turn speed than on temporal variations in turn duration.

To get a qualitative understanding of how turn rate and speed depend on the signal, we plotted them as a function of time (**Fig 4A**). In low frequency signals, turn rate and turn speed appeared strongly modulated by each odor encounter (**Fig 4A** left four panels), as previously seen in experiments that used real odors pulses of low frequency [26]. In contrast, in high frequency signals, turn rate and turn speed responded much more strongly to the first and last pulse of each block, while being only weakly modulated by individual odor pulses within the blocks (**Fig 4A** right four panels), as previously observed in flies navigating complex plumes where odor encounters continuously occurred at high frequency [27]. Closer examination of the data shows a similar temporal novelty response and offset response as we observed for angular speed (**Fig 3A**): turn rate and turn speed spiked at each pulse onset, however for the higher frequencies the responses were stronger at the onset of the ON block than for the subsequent odor encounters. At high intermittencies there was also an off-response at the offset of the ON block (**Figs 4A,** grey **and S9 and S10,** grey). We conclude that turn dynamics are mainly controlled by signal-driven modulations of the turn rate and turn speed, which exhibit both a novelty-response and offset-response.

To model turn rate and turn speed at signal offset, we defined an intermittency-dependent offset response *OFF(t)* analogous to the OFF response reported by [26], who used real odors to stimulate flies in a setup similar to ours. The *OFF(t)* function computes the difference between two integrative filters which decay at different timescales, one long and one short, producing a transient spike after long duration encounters or higher intermittency signals (**Fig 4B**)

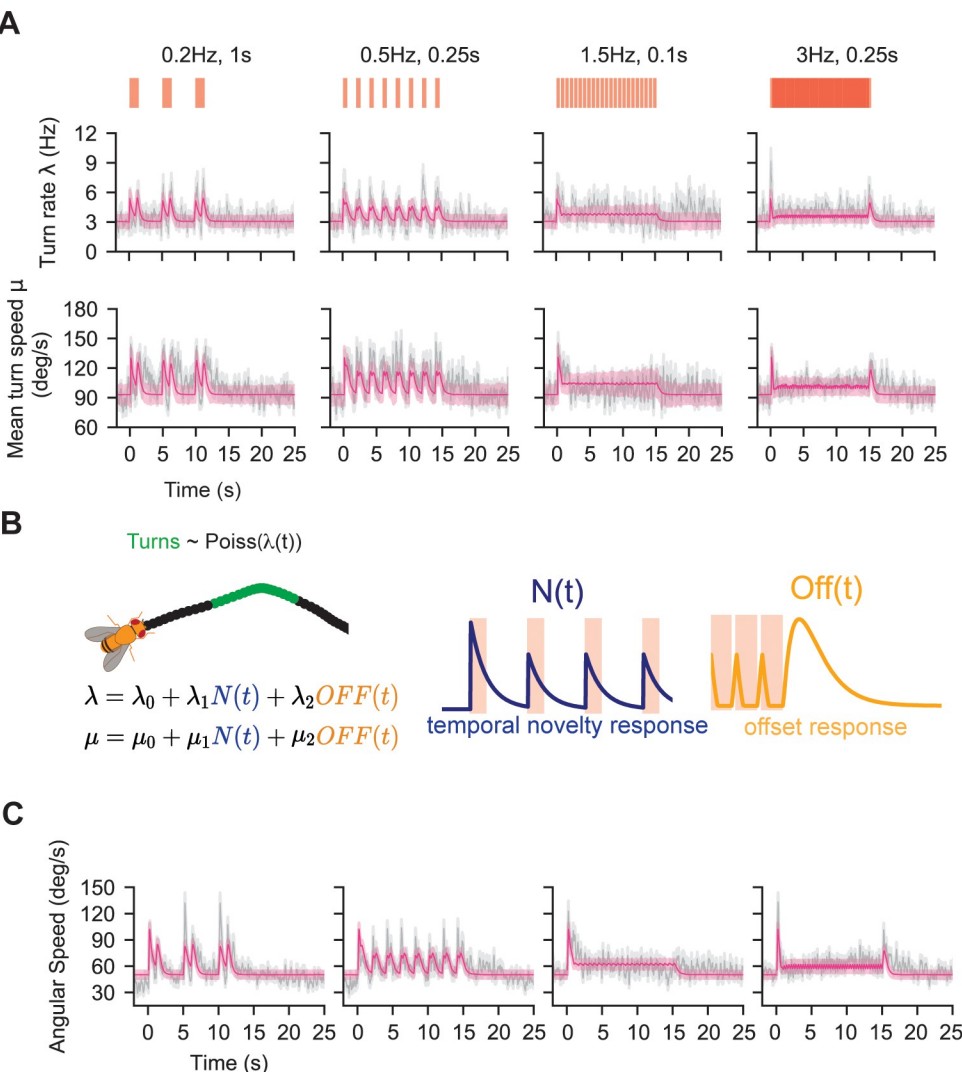

**Fig 4. Temporal novelty detection and offset response together can predict turn rate, angular speed given turning, and angular speed dynamics. A.** Population mean turn rate (top) and mean turn speed (bottom) from four of the 45 odor environments: 0.2 Hz 1 s, 0.5 Hz 0.25 s, 1.5 Hz 0.1s s, 3 Hz 0.25 s. Grey shading: mean ± SEM. We used a 0.25s sliding window shifted by 1 frame (0.016s) to obtain turn rate and turn speed (see Materials and Methods). Pink line: mean predicted turn rate over time ($\lambda(t)$, top) and mean predicted turn speed over time (bottom). Parameters of Eqs [1–3] were estimated using Maximum Likelihood Estimation (Materials and Methods). Pink shading: standard deviation obtained from repeated simulation of model prediction (Materials and Methods and **S9 Fig** caption). Red bars: fictive odor pulses. We had 8–103 turns per frame across all 45 odor environments. **B.** Model of turn dynamics. Flies initiate discrete turn events (green) with a defined mean turn speed. Turn initiations are modelled as an inhomogeneous Poisson process with rate $\lambda(t)$ calculated as a linear combination of a baseline turn rate, a temporal novelty detector (N(t), blue), and an offset detector (Off(t), yellow) (Eq 2). Mean turn speed was modelled similarly (Eq 3). Fits and their errors are shown in pink in **A. C.** Grey: Mean angular speed of the flies. Grey shade: mean ± SEM. Pink: model prediction for the mean, pink shade: mean ± SEM.

(Materials and Methods). This makes it a good candidate for modeling the observed offset behavior in turn rate and angular speed.

For the frequency-dependent novelty detection, we defined the function *N(t)* that would spike at pulse onsets and then decay until the next pulse onset. The height of a spike increases

with the time since the last pulse onset (i.e. when a pulse has more "novelty" to it):

$$N(t) = A(t)e^{-\frac{t-t_L}{\tau_d}} \tag{1}$$

where $t_L$ is the time of the latest pulse onset, and $\tau_d$ is the decay timescale of the response to individual odor pulse. $A(t)$ controls the height of the response to each pulse. It is maximal (= 1) for the initial odor encounter but decays for successive encounters that occur within a novelty timescale $\tau_N$ (see Materials and Methods). Odor encounters that occur after a time much larger than $\tau_N$ are treated as novel signals and elicit maximal response again (**Fig 4B**).

Given these two response functions, *OFF*(*t*) and *N*(*t*), we attempted to capture the dynamics of both the turn rate and turn speed by simple linear combination. We modeled the turn rate $\lambda$(*t*) as

$$\lambda(t) = \lambda_0 + \lambda_1 N(t) + \lambda_2 OFF(t) \tag{2}$$

and the mean turn speed $\mu(t)$ as

$$\mu(t) = \mu_0 + \mu_1 N(t) + \mu_2 OFF(t) \tag{3}$$

where $\lambda_i$ and $\mu_i$ for $i$ = 0, 1, 2 are constant parameters. We note that turn duration was weakly modulated by the signal statistics (**S11 Fig**), but the modulation was much smaller compared to modulations in turn rate and turn speed. Thus, for simplicity, we treated turn duration as exponentially-distributed with fixed parameters determined from data (**Fig 3D**). We first estimated the parameters for $\lambda(t)$ using maximum likelihood estimation (see Materials and Methods), carrying out the estimation by pooling data from all 45 stimulus environments. This fixed $\lambda_0$, $\lambda_1$, and $\lambda_2$, as well as the timescales involved in the *N*(*t*) and *OFF*(*t*) responses (Materials and Methods and Table 1). Then, holding the timescales fixed, we estimated the $\mu$ coefficients, again fitting to the data pooled from all 45 environments (Materials and Methods and Table 1). Our model captured both turn rate and turn speed well, albeit slightly underestimating both at lower frequencies (**Figs 4A**, pink and **S9 and S10,** pink).

Up to this point, our model captures how the stimulus modulates the rate and magnitude of discrete turn events. These two aspects, along with turn duration, which we held fixed, should predict angular speed across environments with diverse signal frequency and intermittency. To test this, we simulated virtual agents enacting our dynamic model. Agents executed turns via an inhomogeneous Poisson process following Eq 2. The mean angular speed of each turn was sampled from a Gamma distribution with signal-dependent mean (Eq 3 and Materials and Methods). The turn duration was sampled from a fixed exponential distribution (Materials and Methods). In our dataset there are at most 240 fly trajectories at any given time. Therefore, we simulated 240 agents in each of the 45 environments to get a population-averaged trace, and then repeated this process 10,000 times to get an estimate of the model-predicted mean and error. To quantify model accuracy, we calculated the ratio of the root-mean-square error of the model fit to the data standard deviation (NR score) across all environments [41,42]. An NR score of <1 indicates a model prediction within the noise of the data. The NR score across all 45 environments was 0.16, indicating that the model recapitulates the dynamics of the fly angular speed well across experiments, albeit with some underestimation at the lowest frequencies (see **Fig 4C**; predictions for all 45 environments are shown and discussed in more detail below).

The model reproduces two important aspects of the turning dynamics and its dependency on signal frequency and intermittency: a varying turn rate at the onset and offset of longer odor encounters, and a roughly constant turn rate when the frequency of encounters is high

**Table 1. All Fit Parameters.**

| Parameter | Explanation | Estimation Bounds | Fit value |
|---|---|---|---|
| $\tau_N$ | novelty detection timescale | 0.01-1200s | 2.04 s |
| $\tau_{ND}$ | novelty response decay timescale | 0.01-1200s | 0.55 s |
| $\tau_{fast}$ | off response fast timescale | 0.01-1200s | 0.19 s |
| $\tau_{slow}$ | off response slow timescale | 0.01-1200s | 0.22 s |
| $\lambda_0$ | base turn rate | -infinity to infinity | 3.06 /s |
| $\lambda_1$ | turn rate novelty response coefficient | -infinity to infinity | 2.80 /s |
| $\lambda_2$ | turn rate off response coefficient | -infinity to infinity | 45.14 /s |
| $\mu_0$ | base mean turn speed* (see below) | -infinity to infinity | 68.1 deg/s |
| $\mu_1$ | mean turn speed novelty response coefficient | -infinity to infinity | 47.1 deg/s |
| $\mu_2$ | mean turn speed off response | -infinity to infinity | 582.0 deg/s |
| $\tau_{dur}$ | turn duration timescale* (see below) | 0.01-1200s | 0.18s |
| $a_0$ | upwind bias baseline shift | -infinity to infinity | -0.49 |
| $g$ for $I(t)$ | upwind bias response gain using intermittency-sensing model | 0–16 | 12.6 |
| $\tau_I$ | intermittency-sensing model integration timescale | 0.01-5s | 0.04s |
| $g$ for $F(t)$ | upwind bias response gain using frequency-sensing model | 0–16 | 9.3 |
| $\tau_F$ | frequency-sensing model integration timescale | 0.05-5s | 0.08s |
| $g_I$ for $H(t)$ | intermittency sensor gain for dual frequency and intermittency sensing model | 0–4 | 2.7 |
| $g_F$ for $H(t)$ | frequency sensor gain for dual frequency and intermittency sensing model | 0–4 | 3.2 |
| $\tau_H$ | dual frequency and intermittency sensing model integration timescale | 0.05-5s | 0.1s |
| $g$ for $R(t)$ | upwind bias response gain using two-timescale integrator | 0.15–3 | 1.5 |
| $\tau_g$ | upwind bias rise timescale for two-timescale integrator | 0.01-5s | 0.01s** |
| $\tau_d$ | upwind bias decay timescale for two-timescale integrator | 0.05-5s | 0.97 s |

*Note that to fit turn speed and turn duration, we first subtracted the minimum turn speed (25 deg/s) and minimum turn duration (0.18s) set by our thresholding method. These values ($\mu_0$ and $\tau_{dur}$) represent the addition to the minimum threshold values set, that were fit. Thus in the absence of signal the mean turn speed is actually 25 + 68.1 = 93.1 deg/s and the mean turn duration is 0.18 + 0.18 = 0.36s: the mean turn duration beyond the minimum duration coincidentally was also fit to 0.18s.

**Note that the found value of 10ms for $\tau_g$ was on the edge of the minimization bounds. We did not probe shorter timescales as the time-resolution of our data was ~16ms. However we verified that even for an instantaneous rise timescale the cost was higher than the cost for 10ms but within 3% of its value. The same was true for a $\tau_g$ of 20ms. For values greater than 20ms the cost started to grow more significantly. Hence we concluded that the rise timescale was between 0-20ms.

(after the initial spike), both of which have been observed experimentally in separate paradigms investigating these distinct odor environments [26,27].

## Upwind bias responds to odor signal with two timescales: a fast rise time and a slow decay

Up until now, we are able to describe fly angular speed dynamics well, through a dynamic turn rate and turn speed. In order to describe fly orientation, we must also understand the direction of these turns, controlled by the upwind bias–the probability that a given turn is upwind [27]. To illustrate how the upwind bias depends on signal, we plotted it in time (Materials and Methods), finding that in general it was dynamic and high during the ON block, but otherwise slightly below 0.5 (**Figs 5A**, grey and **S12**, grey). Unlike the turn rate and turn speed, upwind bias also depended on fly orientation, and was largest for crosswind-facing flies (**Fig 5B**).

Following our previous work [27], we model upwind bias $B(t)$ as a sigmoid (**Fig 5C**):

$$B(t) = \frac{1}{1 + \exp[-(a_0 + g \cdot u(t)) \cdot \sin^2\theta]} \qquad (4)$$

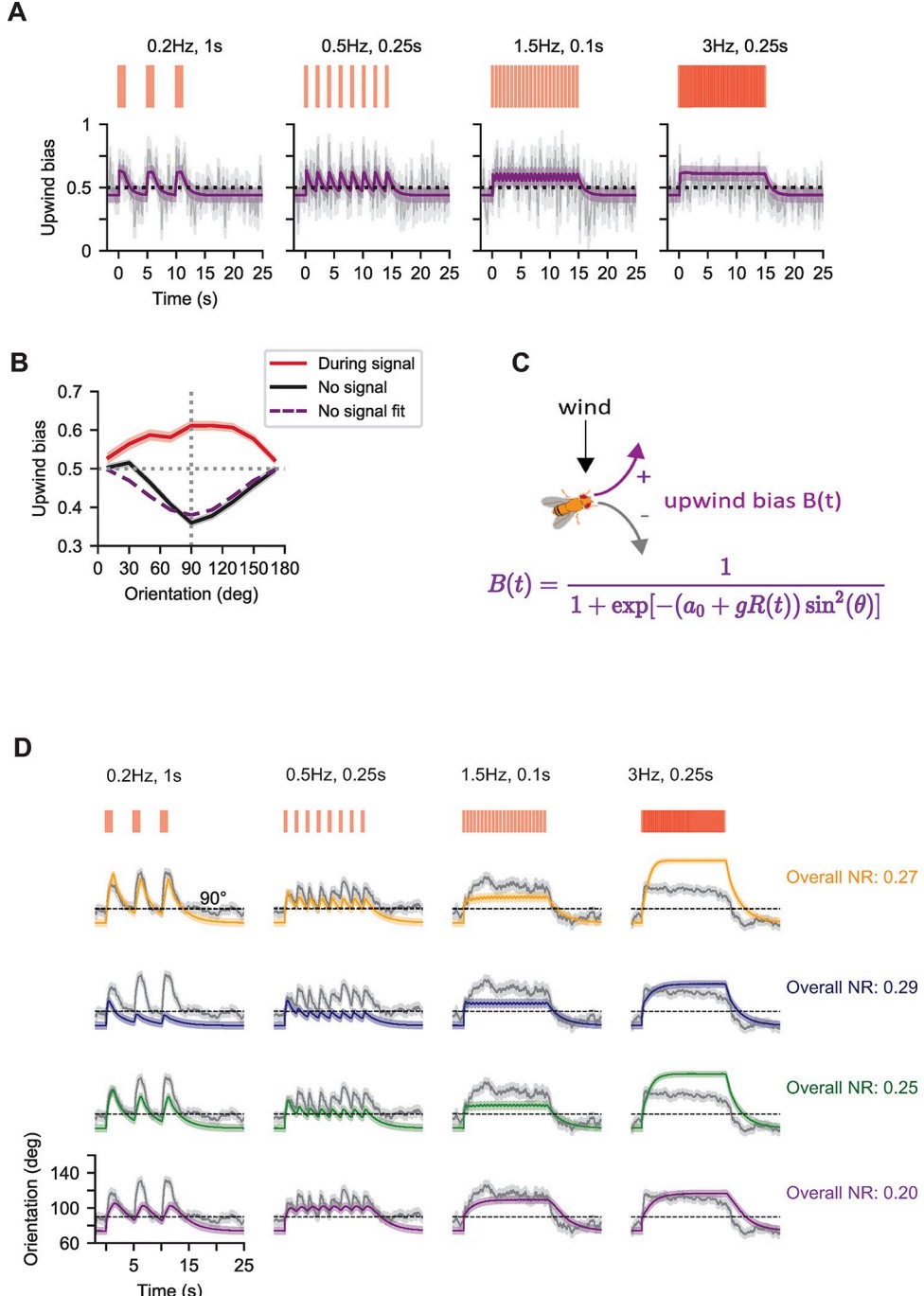

**Fig 5. Comparison of flies' orientations with simple models suggests that upwind bias is modulated over two timescales. A.** Probability to turn upwind given turning (i.e. upwind bias) as a function of time for the same 4 environments as in **Fig 3A** (grey) and predictions from a two-timescale integrator with an instantaneous rise timescale (purple). Experimental upwind bias is estimated using a 0.25s sliding window (Materials and Methods). **B.** Upwind bias vs orientation at two different signal times, across all 45 fictive odor environments. 0 degrees denotes downwind facing flies; 180 degrees denotes upwind facing flies. During the signal (times when the fictive odor flashes were on), turns tend to be oriented upwind with crosswind-facing flies showing the strongest bias (red). When there is no signal present (20s-30s, i.e. last 10s of the OFF block), flies tend to turn downwind and the bias is strongest for crosswind-facing flies (black). Dashed purple: model fit for upwind bias vs orientation when there is no signal. **C.** A model for turn bias suggested by the data. We use a sigmoidal form as in [27,31] where the likelihood to turn upwind given turning increases with $R(t)$, the two-timescale integrating response with an instantaneous rise timescale and finite decay timescale, while the $\sin^2 \theta$ factor ensures that the bias is maximal for crosswind orientations and suppressed at

upwind or downwind orientations. **D.** Comparison of prediction from the four signal processing models in the same four odor environments as in panel A. Grey: mean fly orientation. Grey shading: the standard mean error of the experimental response data. Color: best prediction for each model (Parameters in Materials and Methods and Table 1). Yellow: best prediction of intermittency-sensing model $I(t)$, blue: best predition of frequency-sensing model $F(t)$, green: best prediction of dual intermittency and frequency sensing model $H(t)$, purple: best prediction of two-timescale integrating model $R(t)$. Black dashed line indicates crosswind orientation at 90˚. Scales for time and orientation are given by the horizontal and vertical black solid bars respectively. NR scores are calculated using all 45 environments.

where $a_0$ represents a baseline bias (i.e. when no signal is present), and the $\sin^2 \theta$ term ensures that the bias is maximal at crosswind angles (**Fig 5C**). $u(t)$ is the output of a signal processing model and $g$ is a gain factor controlling how much the signal affects upwind bias. To best capture the upwind bias, we consider four simple signal processing models $u(t) = I(t), F(t), H(t), R(t)$ hereafter called: intermittency sensing $I(t)$, frequency sensing $F(t)$, dual-frequency-intermittency sensing $H(t)$ and two-timescale integrator $R(t)$. The intermittency sensing model exponentially filters the binary signal $S(t)$ with timescale $\tau$:

$$I(t) = \int_0^t \frac{1}{\tau} e^{-\frac{(t-t')}{\tau_I}} S(t') dt' \tag{5}$$

By construction, $I(t)$ responds uniquely to signal intermittency [31]. The frequency sensing model $F(t)$ was proposed in [27]. Here, the duration of the signal is ignored, and the signal is converted to a time-series $w(t)$ of delta function spikes at the onset of each odor encounter. The onsets are exponentially filtered to produce the output, which is effectively a running estimate of odor encounter frequency:

$$F(t) = \int_0^t e^{-\frac{(t-t')}{\tau_F}} w(t') dt' \tag{6}$$

The third model examined was a dual frequency and intermittency sensing model $H(t)$, outlined in [31]. This model linearly combines $I(t)$ and $F(t)$, but the contributions of $F(t)$ and $I(t)$ are independently weighted:

$$H(t) = g_I \cdot I(t) + g_2 \cdot F(t) \tag{7}$$

Here $g_I$ and $g_F$ are gain factors. In this case the gain $g$ in Eq [4] is set to be 1 and the same timescale is assumed for $I$ and $F$. Finally, in the two-timescale integrator model, the response $R(t)$ adapted to the signal with one timescale $\tau_g$ when the signal turned on, but another timescale $\tau_d$ when the signal was lost. This is expressed mathematically as:

$$\frac{dR}{dt} = \frac{1}{\tau_g} \cdot (S(t) - R(t)) \tag{8}$$

when the signal is on and

$$\frac{dR}{dt} = \frac{-1}{\tau_d} \cdot R(t) \tag{9}$$

when no signal is present. This model always responds to signal intermittency, but also responds to frequency independently, up to frequencies of $\frac{1}{\tau_d}$, provided $\tau_g \ll \tau_d$ (see Materials and Methods). Eqs [4–9] define 4 alternative models for the upwind bias. Together with the turn dynamics model described in the previous section, this provides us with 4 alternative models to predict fly orientation dynamics.

To find out which of these 4 models best describes fly behavior, we fit all of them to data. To constrain $a_0$ we took advantage of the fact that the upwind bias returns to baseline within a couple of seconds following the offset of the ON block (**Figs 5A**, grey and **S12,** grey). Accordingly, we estimate $a_0$ by using the last 10s of the OFF block (**Fig 5D**, black) to fit Eq [4] with $u(t) = 0$ (**Fig 5A**, purple). To estimate the remaining parameters we simulated stochastic agents. Turn initiation, speed and duration were simulated as explained above using the best fit parameter values extracted from the analysis in the previous section. Turn bias parameters were estimated using Eq [4] to generate a stochastic turn direction for each turn executed by the agents. For each environment, 240 sample trajectories were generated. We constrained the turn bias parameters by minimizing the mean squared error between the mean orientation of agents and flies (Materials and Methods).

We found that the two-timescale integrator model $R(t)$ best fit the data across all environments **(Fig 5D)**. The intermittency sensing model ($I(t)$, overall NR = 0.27, optimal parameters $\tau_I = 0.04$s, $g = 12.6$) predicted well the response for long encounters or lower frequencies, but underestimated responses at higher frequencies or lower durations, and overestimated the response at the highest intermittencies (**Fig 5D**, yellow). Conversely, the frequency sensing model ($F(t)$, overall NR = 0.29; best fit parameters $\tau_F = 0.08$s, $g = 9.3$) exhibited the opposite trend: satisfactory fits for frequencies $> \sim 1.5$ Hz, but clear underestimates for lower frequencies (**Fig 5D**, blue). This suggested that a simple sum of these models might resolve these individual failure modes. Indeed, the dual-frequency-intermittency model was more accurate overall ($H(t)$, NR = 0.25), and with optimally fit gains $g_I = 2.7$, $g_F = 3.2$ and timescale $\tau_H = 0.1$s, captured the mean and dynamics the orientation response across a range of odor environments (**Fig 5D**, green). Still, it underestimated the response at low frequencies $< 1.5$ Hz. The two-timescale integrator model, however, predicted the mean orientation responses across all panels better ($R(t)$, NR = 0.20) than the than the dual-frequency-intermittency model. The optimally fit rise time for the response at signal onset was almost instantaneous ($\tau_g = 0.01$s— Materials and Methods for details about this value), whereas the decay timescale was much longer ($\tau_d = 1$s). We also verified that the two-timescale integrator models with best-fit parameter values reproduces upwind bias in the data (**Figs 5A**, purple and **S12**, purple).

We conclude that models that combine frequency and intermittency sensing to determine upwind bias fit the data better than single-sensor models [31]. However, a linear combination of the frequency and intermittency sensors is not sufficient. The data is better reproduced by sensor that responds to these features through integrating the odor signal over two different timescales.

## A single model captures general trends in angular speed and orientation across a broad spectrum of temporally diverse fictive odor environments

To better examine the limits of our model, we now plot mean angular speed and orientation predictions on top of the data across all 45 environments. As mentioned above, angular speed is predicted well across most environments (**Fig 6,** NR = 0.16) and the model captures the variation of the turning dynamics with respect to signal frequency and intermittency, recapitulating differences previously seen between experiments that explored different signal parameter regimes [26,27].

The model also captures the general trend in orientation (**Fig 7,** NR = 0.20), albeit less well than for angular speed. This is to be expected given the cumulative effect that errors in the prediction of angular speed and turning bias have on orientation. Maxima in mean fly angular speed and orientation were underestimated at low frequencies, but overall general trends across all 45 panels were captured.

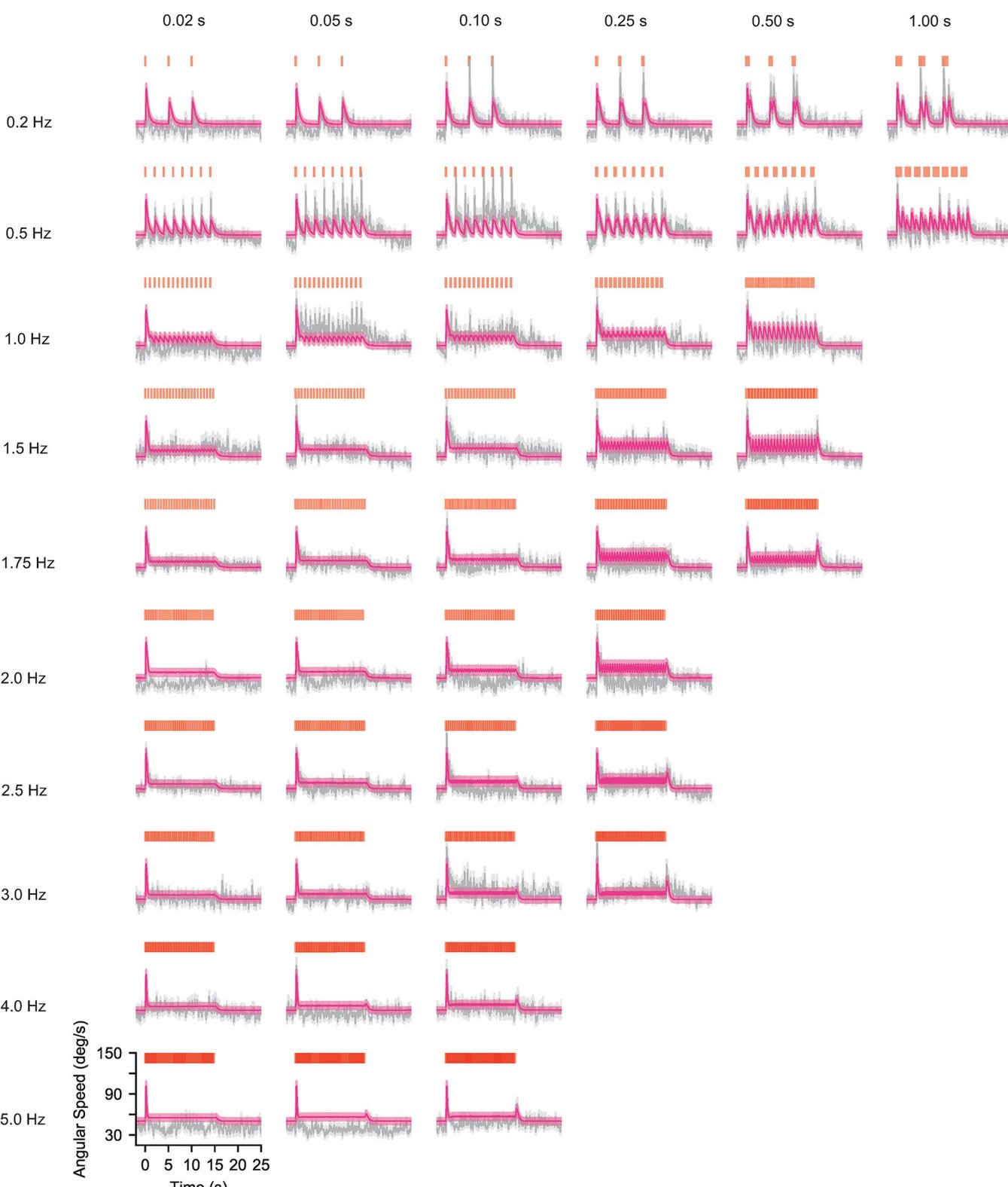

**Fig 6. A single model with fixed parameters captures general trends in angular speed across a spectrum of temporally diverse fictive odor environments.**
Population mean angular speed (grey) and model predictions (pink). Grey shading denotes standard error of the mean while pink shading denotes simulated standard deviation (see Materials and Methods). Trends in population angular speed, which is modelled as the result of dynamic turn rates and mean turn angular speeds (Eqs 2 and 3, respectively), are well captured by the model across all 45 experiments.

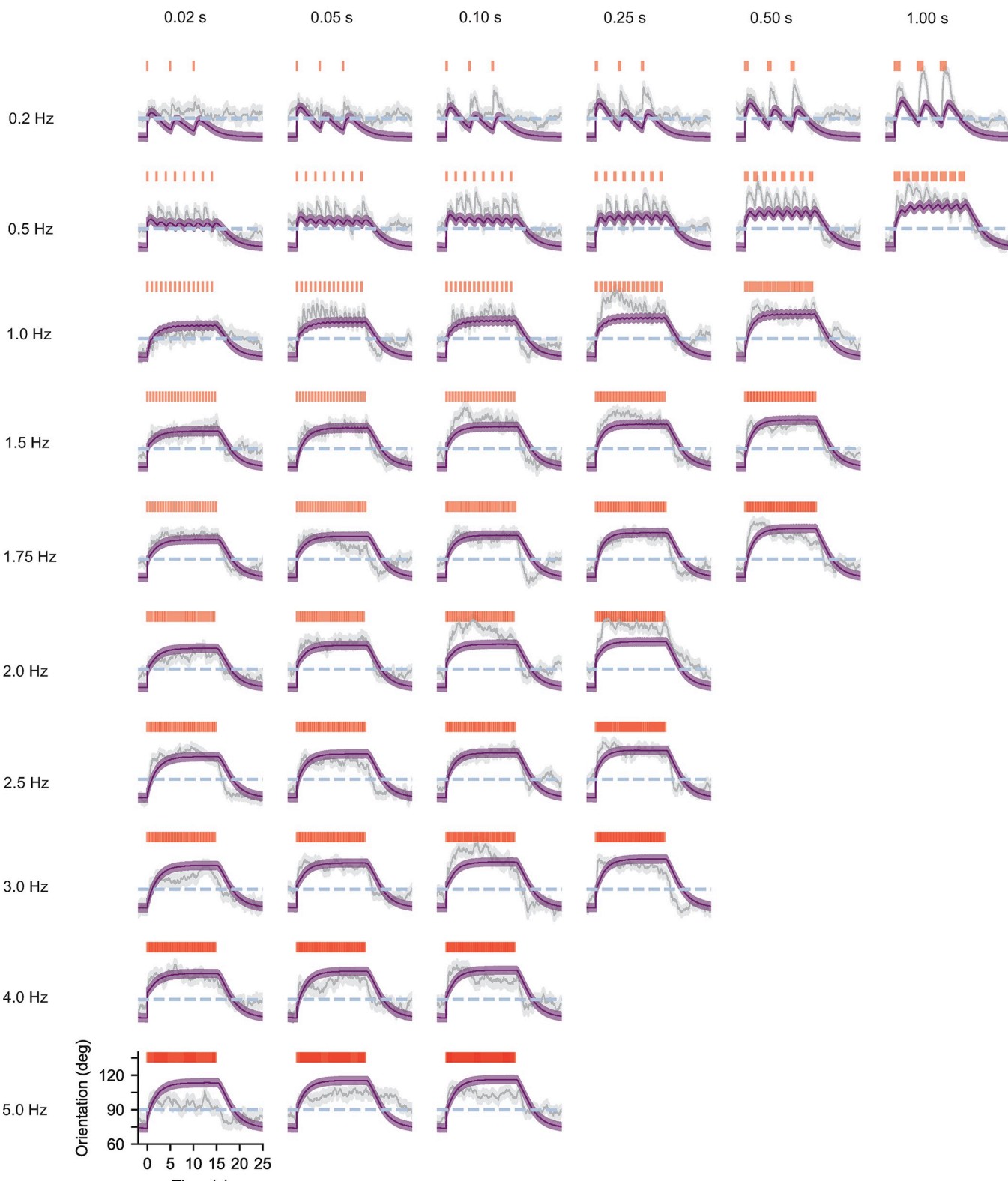

**Fig 7. A single model with fixed parameters captures general trends in orientation across a spectrum of temporally diverse fictive odor environments.**
Population mean orientation (grey) and standard error (grey shading) along with model predictions (purple) and estimated model errors (purple shading). 180 degrees is upwind, 90 degrees is crosswind and 0 degrees is downwind. Dashed blue line indicates 90 degrees. Model orientation is the result of simulated turns

occurring with dynamic rates (Eq 2) and average angular speeds (Eq 3) as well as a dynamic bias towards the upwind direction (Eq 4). We see that general trends and mean values across all 45 different environments are well-approximated by our model. Number of trajectories per time point is the same as in **Fig 2A**.

We conclude that flies navigate diverse temporal statistics by: 1) modulating their turn decisions and turn speed via a frequency-dependent novelty detector and an intermittency-dependent offset detector of odor signals; and 2) biasing the orientation of these turns using a response function that integrates the signal over two timescales, a very fast rise timescale (tens of ms) and a slow decay timescale (seconds).

## Temporal novelty detection can be combined with odor motion sensing to aid navigation in low frequency environments

Finally, we wondered whether the features of our temporally driven model, when combined with other known sensory modalities, enhance navigation of more complex odor environments. Of the key features of our model mentioned above, the effect of the offset response has been explored previously [26]. Furthermore, we show in Materials and Methods that the two-timescale integrating response increases with both intermittency and frequency. Together with our results from [31] this implies that the two-time scale response would aid in navigation across a range of temporally diverse olfactory environments. However, we were curious to see how the temporal novelty response affects navigation, especially when combined with the newly-discovered odor motion sensing [28].

To address these questions, we first added odor motion sensing to the model described in previous paragraphs. Following [28] agents can sense the wind direction and compute an odor motion direction at each timepoint, using a Hassenstein-Reichardt correlator [43] (Materials and Methods). Agents initiate turns with rates and speeds in accordance with our model (Eqs 2 and 3). If an agent is turning and the odor motion signal is above a set threshold at the time of turn initiation, then the agent turns towards the vector sum of the upwind and against-motion directions (**Fig 8A** and Materials and Methods). If the odor motion signal is below the threshold then the turn is biased as in Fig 5C.

The odor plume was generated as described in [28] following the approach detailed in [44]: odor packets are released from a source at a Poisson rate and advected downwind by a constant velocity while simultaneously performing a correlated random walk in the perpendicular direction to model turbulent dispersion [45]. Odor packets also diffuse around their center of mass according to molecular diffusion (**Fig 8B** and Materials and Methods). Given that the novelty response is not strong in environments where the frequency is much larger than $\frac{1}{\tau_N} \approx$ 0.5 Hz, we first tested this navigation model in an environment where odor encounter frequency was $\leq$0.5 Hz in the bulk of the plume (**Fig 8C**, top). We simulated 100,000 agents for each condition. We quantified success by calculating the fraction of agents that reach a 25mm by 25mm target region around the source (**Fig 8B**). To examine how performance depends on the different features of the model we normalize success with that of the full model (**Fig 8D**).

Full model agents were able to get close to the odor source (**Fig 8D**) much more than in a control simulation where no odor signal was presented (**Fig 8D** right most bar). Removing novelty detection by replacing the novelty response $N(t)$ in turn rate or turn speed, or both, by its average across the 100,000 agents, significantly reduced performance by about 20%. However, in agents that did not have odor motion sensing capability, the removal of novelty detection had only a minor effect (**Fig 8D**). This suggests that temporal novelty detection might be most useful for navigation when combined with motion sensing ability.

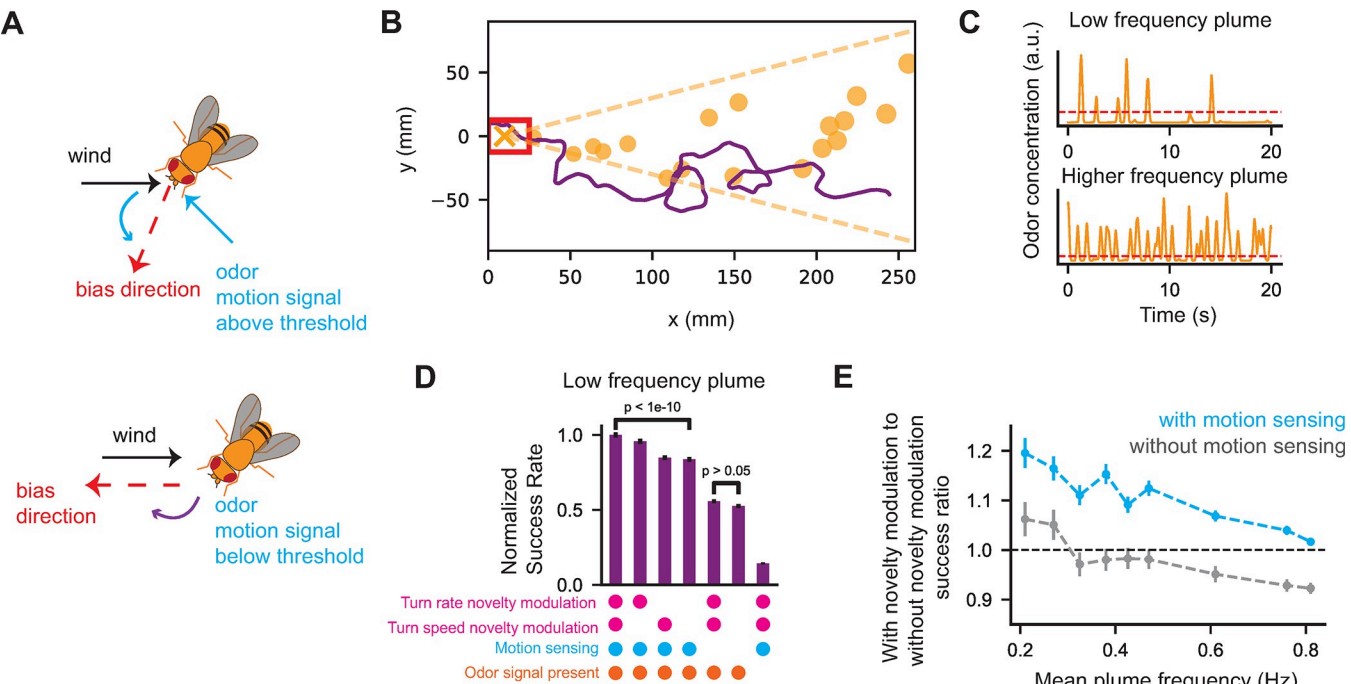

**Fig 8. Novelty detection combined with odor motion sensing improves navigation performance in low frequency environments. A.** Illustration of how odor motion sensing is combined with our model. If a turn is being initiated and motion signal is above threshold, then agents turn against the vector sum of wind and motion directions. Otherwise, agents execute the model summarized in Figs 6 and 7. **B.** Snapshot of an odor plume simulation. Red box denotes success region. Orange X denotes source location. Orange circles denote simulated Gaussian packet locations and sizes. Orange dotted lines show plume envelope, set by the ratio of crosswind speed to downwind speed. Purple curve denotes example successful simulated trajectory. **C.** Example odor concentration time series in two simulated plumes. Dashed red line indicates odor detection threshold. **D.** Normalized fraction of successful simulated agents in the low frequency plume. Simulation conditions (x-axis) are distinguished by which features are modulated by the novelty detection response (see Eqs 2 and 3) and whether motion sensing is present. As a control we also show performance of agents when no odor is present (right-most bar). In the low frequency plume and with motion sensing capability, there is a statistically significant improvement in performance when novelty detection is present. 100,000 agents were simulated for each condition. Significance calculated using a 2-proportion Z-test. Error bars obtained by bootstrapping (Materials and Methods). **E.** Ratio of navigation success rate of agents with turn rate and turn speed novelty modulation to agents without novelty modulation, in different plumes. Plumes are distinguished by average encounter frequency. Blue (grey) curve is for agents with (without) odor motion sensing. 100,000 agents were simulated in each plume for each condition. Error bars for D and E obtained by bootstrapping (Materials and Methods).

Finally, to test our hypothesis that novelty modulation is more helpful in lower frequency environments, we simulated agents navigating plumes with different encounter frequencies (Materials and Methods), ranging from 0.2 Hz averaged across the plume in the low frequency plume (Fig 8B, top) to 0.8 Hz (Fig 8B, bottom). For each plume we simulated 100,000 agents with and without novelty modulation, replacing as above $N(t)$ for both turn rate and turn speed modulation by its average value across the agents with novelty modulation. We found that at lower frequencies the success rate of agents with novelty modulation was higher than that of agents without novelty modulation. But as expected, the benefit of temporal novelty detection dropped as encounter frequency increased (Fig 8E, blue). Furthermore, across all environments, the relative value of novelty detection was significantly diminished for agents that did not have odor motion-sensing capability (Fig 8E, grey). Together, these findings suggest that temporal novelty detection aids olfactory navigation in environments where encounter frequency is less than $1/\tau_N$ (which using our experimentally fit value comes out to 0.5 Hz), provided agents are also able to sense odor motion.

## Discussion

It is well-known that animals from crabs [46] to moths [23] and *Drosophila* [25,47] use various temporal features of olfactory stimuli to modulate navigation. Previous studies in walking flies have shown that turns can be modulated by the frequency of odor encounters in complex plumes, and by encounter duration in low frequency environments [26,27,31]. Here we carefully examined the transition between these two regimes. To isolate temporal features from spatial information such as the spatial structure of odor encounters, local odor gradients, and turbulent wind structure, we used optogenetics. This allowed us to probe a broad range of odor frequencies and durations.

A key finding of this study is that a model incorporating both a frequency-dependent novelty response (this study) and a previously observed intermittency-driven offset response [26] can successfully describe the dynamics of turns across the spectrum of temporally diverse environments studied. This single model predicts that in environments with high odor intermittency and low frequency, the turn rate is dynamic and spikes at onset and offset of each odor encounter, as seen in [26], whereas in environments of high frequency odor encounters, the turn rate remains roughly constant, as seen in [27], after the initial response to the first encounter. This temporal novelty response was mostly not observed in [27] (although it was present in Fig 2 and Supplement 1B of that study). In our current study, such a feature is likely highlighted due to all flies receiving identical and step-like odor stimuli simultaneously.

The novelty response reveals that after ~2s of no stimulus a fly is very likely to execute a large turn in response to a new odorant encounter, whereas more frequent encounters are not as likely to trigger such a deterministic response. In turbulent odor plumes, times between encounters (blank times) are power-law distributed and odor packets tend to arrive in clumps [2,3,11,13]. Thus, within natural plumes a fly may not experience any odor packet for an extended period of time. Furthermore, our temporal novelty response suggests that flies may respond differently to the beginning of a clump than to fluctuations within the clump. It would thus be interesting to compare this novelty timescale with the distribution of blank times and clumps durations in natural plumes. Flying flies also experience very different signal statistics from walking flies, which begs the question of whether this novelty timescale is the same in flying flies. The observed spikes in turn rate and turn speed are also transient, decaying with a timescale of about 0.5s. Probing the basis of this novelty response and how these observed timescales emerge from the neural circuitry could be a fruitful avenue for future study.

Our final simulations show that temporal novelty response enhances navigation in lower frequency ($\leq \frac{1}{\tau_N} \approx 0.5$ Hz) environments, particularly when combined with odor motion sensing. This suggests that the novelty response allows agents to capitalize on odor motion signals and turn against odor motion in low frequency environments, which helps localizing the plume center-line [28]. However, odor motion signals are also present and useful in much higher frequency environments [28]. Why would flies not turn against the direction of each odor encounter in higher frequency environments as well? One possible answer is that turning at higher and higher frequencies means that it would take longer for flies explore their environment, as in general the diffusion coefficient of such an agent is inversely proportional to its reorientation rate [48]. It is also possible that in higher frequency environments, flies can average over several hits to get more reliable estimates of signal statistics or motion and hence do not need to turn in response to each hit. In general, our findings suggest that it would be interesting to investigate what temporal novelty timescales are optimal for navigation in different environments and how this depends on the odor motion signals in the environment as well.

Furthermore, our model was simplistic in that it assumed that agents weight the upwind and against-motion direction equally when a sufficiently strong motion signal is present. It

would be interesting to investigate more carefully the relative weighting of the upwind and against-motion directions of turning *Drosophila* and how this might depend on the temporal statistics of the odor plume.

In addition to the modulation of turn rate and turn speed, another important finding of this study is that the likelihood for a turn to be oriented upwind increases with a very fast time-scale (fit to be roughly 10ms) at signal onset and decays with a slower timescale of roughly 1s. We show (Materials and Methods) that as a result, upwind bias increases independently with both signal frequency and intermittency, thus allowing for a sustained upwind bias across environments. The response to both frequency and intermittency is largely consistent with previous findings that intermittency dominates upwind motion in high-duration, low frequency environments whereas frequency dominates in low-duration, high frequency environments [26,27,31].

The multiple timescale integration observed here in fly behavior is within the range of the fast and precise processing capabilities of the *Drosophila* olfactory circuit. *Drosophila* ORNs process signal as fast as 100 Hz [49]. This information is preserved downstream where 2nd-order projection neurons (PNs) encode a broad range of signal frequencies via multiple post-synaptic currents [37,50–52]. Deeper downstream computations in the fly brain integrate odor information with wind direction [36] and drive motor actions [53], enabling rapid behavioral responses (on timescales of ~50ms) [54].

A recurring theme in *Drosophila* temporal odor processing, both in behavior and circuitry, is the importance of two distinct timescales. At the first processing relay, ORNs synapse onto PNs with two kinetically distinct fast and slow postsynaptic currents which promote a wide range of frequency transmission [50] and promote robust navigation of simulated flies across environments with diverse temporal statistics [31]. Our model of fly turning exhibits similar fast and slow timescales: the turning bias increases rapidly ~10ms at odor onset but decays slowly ~1s at odor offset. Moreover, our mathematical analysis (Materials and Methods) shows that it is this asymmetry-a fast rise and slower decay-that enables a two-timescale inte-grator response to increase with intermittency and frequency independently.

*Drosophila* ORNs adapt their activity to both the mean and variance of fluctuating odor sti-muli (adults: [42,55–57], larvae: [58]) which aids preservation of both response dynamics [42] and odor encounter timing in ORN spiking [55,59]. Here we used optogenetics to drive behav-ior, thus bypassing the part of the ORN adaptation dynamics that takes place upstream of the firing machinery [55,57]. In a previous paper that used the exact same experimental setup and light intensity, we verified that the type of stimuli used here drives ORN responses within their physiological range and that fly behavior resembles that in real odor plumes [28].

While most parameters of the model are fit using Maximum Likelihood Estimation with a likelihood function that uses data from individual time points and trajectories (Materials and Methods), we only verify our model's predictions against the average behavior of flies. How-ever, there is considerable individual variability between fruit flies, arising from factors such as genetic differences and varied internal states (e.g. hunger levels) [60]. It would be interesting to study to what extent the features and parameters of our model vary between individuals and how they are affected by these factors. This would require measuring individual flies for much longer than the individual trajectories we could obtain here.

Other aspects of olfactory navigation have not been considered in this study. As we wanted to focus solely on orientation dynamics, we did not factor changes in ground speed or transi-tions between stops and walk bouts into our predictive models, although these locomotory behaviors are known to be modulated by odor encounter timing and duration [26,27,34]. In our dataset, the walking speed exhibited a small modulation around a mean of 10 mm/s with drops at the offset of higher intermittency signals (Panel A in **S13 Fig**), consistent with

previous studies [26,27,34]. The stop-to-walk transition rate also showed modulation (Panel B in **S13 Fig**). Interestingly, the rate also exhibited temporal novelty detection (weaker than for orientation dynamics, see **S9 Fig**), spiking at each encounter in low frequency environments and notably at the onset of ON blocks in higher frequency environments, something we had not observed in (Demir et al., 2020). Modulation of the walk-to-stop rate was weaker. There are several possible explanations for these differences: in (Demir et al., 2020) odor packets had a moving spatial structure, which conveys directional information [28], and their borders were not as abrupt as the step function in light intensity we use here. Additionally, here all ORNs were stimulated.

We acknowledge that this paradigm creates a simplistic odor landscape in which other sensory inputs such as visual cues are removed, which when present can improve navigation success [61,62]. Moreover, any information available to the fly from bilateral sensing was removed due to the spatially uniform signal. Doing so was important to isolate odor timing since insects respond to timing differences across antennae [63], use them to detect odor motion [28] and respond to bilateral concentration differences [9,64]. Since our fictive odor signal activated all Orco-expressing ORNs, which is known to drive a broadly attractive behavior [34], we did not examine the effect of odor identity or valence on turning dynamics [65,66]. Mosquitos and fruit flies, for example, will go upwind in the presence of fluctuating carbon dioxide signals but not when the signal is homogeneous [67,68]. Conversely, mosquitoes will go upwind in the presence of homogenous skin odors but not filamentous skin odors [67]. Moreover, appetitive and aversive memories in flies have been shown to modulate valence and upwind bias [69]. It would thus be interesting to investigate how our observed behaviors and timescales might differ for specific odors or specific appetitive or aversive memories.

We additionally did not investigate any potential effects of flies learning the structure of the odor scene during navigation [70,71], as well as potential collective behavior that could improve odor environment recognition [72].

We have demonstrated that processing odor signals over multiple timescales allows for temporally driven navigational behaviors across diverse environments. Further extensions of this work include investigating the neural bases for these different timescales, as well as how temporal information from individual odor encounters, combined with the overall spatial structure of an odor scene, can be exploited for successful navigation.

## Materials and methods

### Flies/Handling

All fly genotypes used were reared at 25˚C and 60% humidity on a 12 hr/12 hr light-dark cycle in plastic vials containing 10 mL standard glucose-cornmeal medium (Archon Scientific, NC). All flies used in experiments were female, aged 3–10 days old.

To obtain our experimental genotype, we crossed *w;gmr-hid;20X-UAS-CsChrimson* (GMUCR) males with *w;+;Orco-GAL4* (117) virgin females (F1: *w; +/gmr-hid; Orco-GAL4/ 20X-UAS-CsChrimson*). Adults were removed from vials after 3 days, and the F1 females were collected 1–3 days after eclosion. All F1 flies contained a copy of *gmr-hid*, which drives apoptosis [73] in the developing eye in *Drosophila* [74] and are thus blind, and expressed the channelrhodopsin *Chrimson* in their *Orco*-expressing olfactory receptors. 20–30 females were starved 72 hours prior to the experiment in empty plastic vials containing water-soaked cotton plugs at the bottom and top. 24 hours before the experiment, flies were fed 1 mM all trans-Retinal (ATR) (MilliporeSigma) dissolved in water. The vials were covered in foil for these last 24

hours to avoid ATR degradation. For control experiments without ATR (**Figs 1D** and **S2**), flies were instead given 1 mM deionized water.

## Behavioral apparatus

The fly walking arena in this study is identical to that used in [28], based on [27]. The arena was 270 mm x 170 mm x 10 mm (length x width x height). The top and bottom surfaces were made of glass, and walls were acrylic. A plastic mesh was placed downstream of the airflow to prevent flies from escaping, near to which flies were aspirated into the arena through a sealable hole. The arena was illuminated using 850 nm IR LED strips (Waveform Lighting) placed parallel to the sidewalls. Note that although the experimental line is blind, two of the control lines (*Canton-S* and GAL4 parent) are not blind, thus we additionally shone green light using an LED (Luxeon Rebel LED 530 nm) throughout the arena to flood the visual response to simplify comparisons. All other light sources were removed.

Dry air (Airgas) was passed into the arena through a stack of heavy duty plastic coffee stirrers (Mr. Coffee) to present laminarized wind with a flow rate at 150 mm/s. In all experiments, laminar wind was used. To present complex wind within the arena for wind control experiments (**S1 and S3** **Figs**) airflows perpendicular to the laminar flow either side of the laminar mesh were alternately turned on with 100 ms correlation time to perturb the wind structure.

Experiments were recorded at 60 frames per second with a camera (FLIR Grasshopper USB 3.0) with an IR-pass filter. Optogenetic stimuli were delivered using a projector (DLP Light-Crafter 4500) mounted above the arena, with resolution 912 x 1140 pixels, which illuminated the entire walking arena with pixels of size 292 μm (along wind axis) x 292 μm (perpendicular to wind axis). Only the red LED (central wavelength 627 nm) was used throughout this study. All experiments used a 60 Hz stimulus update rate. The projector and camera were aligned by minimizing the least square difference between the two coordinate systems, as described in detail in [28].

## Stimulus protocol

All stimuli were written using custom scripts in Python 3.6.5. All stimuli were delivered to the projector using the Python package PsychoPy, version 2020.2.4.post1.

During signal presentation, the entire arena was illuminated with a spatially uniform pulse of red light ("odor encounter"), presented at the maximum intensity (LED 255). Flies demonstrated similar albeit weaker responses to odor encounters with a lower intensity.

Within one experiment, the stimulus paradigm was repeated four times. Each repeat consisted of an "ON block" and an "OFF block". Odor encounters were presented only during the ON block, which lasted for maximum 15 s. Note that the end of the ON block (i.e. the offset of the last odor encounter) is dependent on the combination of encounter frequency and duration used and thus could be as short as 10.02 s (0.2 Hz, 0.02 s). Any signal that ended after 15 s (i.e. 1.75 Hz 0.25 s, 1.75 Hz 0.5 s, 2.5 Hz, 0.25 s) was terminated at 15 s. The ON block was followed by a 15 s long OFF block, in which no odor was presented. Note that due to the variability in the end of the ON block means that the OFF block could last between 15 s and 19.98 s. Thus each repeat lasted for 30 s, and the entire experiment lasted 120 s. Laminar wind was presented continuously for the span of the experiment unless otherwise stated. Up to 10 experiments were presented to the same set of flies within one session, with a 60 s interval between experiments. The order of the experiments within a session were pseudo-randomized so that two consecutive experiments would not present same stimulus to avoid flies possibly learning from the environment.

## Experimental protocol

Experiments were performed between 08:00 and 12:00 as *Drosophila* activity peaks during this time [75], in a temperature- and humidity-controlled environment (temperature: 22.2˚-C $\pm$ 0.2˚C, humidity: 52.3% $\pm$ 2.7%). Female flies were aspirated into the arena and allowed to acclimatize to the new surroundings and the laminar wind flow for one minute. To ensure that the cross had been successful and that the F1 were healthy and correctly expressing *Chrimson* in their *Orco*-receptors, we presented flies with three parallel static red fictive odor ribbons for 1 minute in laminar airflow. Responsive flies, when encountering the ribbon, tend to turn upwind and weave along the edges of the ribbon towards the expected odor source [27]. Sets of flies that did not show this behavior were discarded. For each combination of encounter frequency and duration investigated, between 6 and 12 videos/experiments were recorded/performed with between 11 and 27 individuals in one session.

## Fly tracking/data acquisition

All tracking scripts were custom written by Nirag Kadakia in Python 3.7.4 and are described in detail in [28].

Briefly, fly centroids were determined using the SimpleBlobDetector function in OpenCV, and assigned to a trajectory identity by matching to other nearby centroids. Centroids that could not be connected to existing trajectories within 30 frames were excluded, and subsequent detected centroids were thus marked as a new trajectory. Orientation was obtained using the *canny* function in *scikit-image* to determine fly "edges", defined between 0 and 360. Measurement noise was removed using a Savitsky-Golay filter (4th order polynomial, window size of 21 frames (0.3 s). The ground velocity in the individual x and y directions were defined by taking the analytical derivative of the fitted polynomials for x and y, and was used to resolve the head and rear of the fly [28]. The angular velocity was determined in the same manner using the orientation. Any potential location bias in the arena due to physical constraints from the stimulus projection were removed by randomly selecting half of the trajectories from each odor environment and flipping the y coordinates and heading along the y axis (axis perpendicular to the airflow). Any trajectory where the fly's mean speed across all the time it was tracked was less than 2mm/s was considered as a non-responsive individual and removed from all further analyses. For Fig 1 only (and S1 and S2 Figs), individual time points where the fly moved less than 2mm/s were additionally removed to ensure equal treatment of data for comparison with data taken from [27]. Our full dataset can be found at [76].

## Defining turns

To define a turn event, we sought to determine a threshold angular speed and minimum duration, above which the reorientation event would be classified as a turn. This method is robust against artificial detections of spurious events that may occur due to measurement fluctuations. However, arbitrarily setting the angular speed threshold too high will neglect large angular changes that likely drive changes in the overall heading. To determine a suitable minimum angular speed threshold for this dataset, we pooled trajectories across all 45 odor environments, and examined how changing the threshold angular speed for a turn event affected the distribution of the angular change for "fixation" events, i.e. the change in orientation for events where the angular speed was below threshold. We set the threshold at values between 5 deg/s and 150 deg/s. For each of the 17 tested thresholds, we obtained the distribution of angular change magnitudes during fixation events, and extracted the 95th percentile to obtain a comparable measure representing the majority of angular changes made (Panel A in **S7 Fig**).

A suitable threshold would have smaller angular changes during fixations, and larger changes during turns. We found that initially, as the threshold increased, smaller angular changes that are likely caused by trivial reorientations were classed as "fixations", and thus would be disregarded as a turn event. Increasing the turn speed threshold beyond 25 deg/s led to much greater changes in orientation during fixation events, (Panel A in **S7 Fig**) (see also **Fig 3B**). Thus we set the minimum angular speed threshold for a turn event at 25 deg/s.

We observed that the distribution of angular change magnitude for events above the 25 deg/s threshold angular speed was bimodal (Panel B in **S7 Fig**). The first peak indicates a proportion of events with small angular changes; the second peak centralized around much larger angular changes. We speculated that the distribution of smaller angular changes could be from very short, sharp changes in orientation, which were potentially artefacts of the tracking to be removed.

We fit a Gaussian mixture to the distribution and found that the standard deviation of the low-mean Gaussian was approximately 4.5˚. A navigating agent is more likely to regulate the duration of its turn, rather than the magnitude of the angular change and thus we instead set a minimum duration for turn events. With a minimum angular speed of 25˚/s, and a minimum angular change of 4.5˚, we get a minimum turn duration of 0.18 s. We removed all above angular speed threshold events with an event duration of less than 0.18 s; the resultant angular change distribution for turn events was no longer bimodal (Panel C in **S7 Fig**).

### Plotting turn quantities as a function of time

To estimate the turn rate as a function of time, at any time point we considered trajectories where flies were in a fixation state or had just transitioned from fixation to turn at that time point. Assuming an inhomogeneous Poisson process, the probability to transition from fixation to turn at a timepoint $t$ is given by $\lambda(t) \cdot \Delta t$, where $\Delta t$ is the time-step resolution of our data, 1/60s. Thus the fraction of all considered trajectories that had just transitioned from fixation to turn, divided by $\Delta t$, provides an estimate of $\lambda(t)$. We smoothed this estimate with a rectangular smoothing window of width 0.25s, sliding the window across each time step, and plotted the results in **Figs 4A**, grey, and **S9**, grey. Errors bars were estimated by bootstrapping the data 500 times at each time point.

We estimated mean turn speed, mean turn duration and upwind bias as a function of time in the following way. We defined a 0.25s wide window and considered all turns that started in this window. For each turn, we then computed its mean angular speed, its duration and whether it was upwind (+1) or downwind (0). We then averaged these quantities to get an estimate of the mean turn speed, mean turn duration or upwind bias, respectively, for that window. As for the turn rate, we slid the window across each time step and plotted our results against the center of the time window. Error bars were estimated by resampling the data with replacement 1000 times at each time point, generating 1000 traces, which we then smoothed and computed the standard deviation for at each time point.

### Modeling fly turning behavior

The *OFF* function is defined as

$$OFF(t) = \max\left(0, I_{slow} - I_{fast}\right) \tag{10}$$

analogously to how it was defined in [26]. Here $I_{slow}$ and $I_{fast}$ are defined as in Eq 5, with characteristic timescales $\tau_{slow} > \tau_{fast}$. At signal offsets, $I_{slow}$ decays slower than $I_{fast}$, so their difference (and thus *OFF*) is positive for some time. At signal onset or presence, $I_{fast}$ rises faster and is

greater in value than $I_{slow}$, thus the max operation ensures that *OFF* is 0. Due to the integrative nature of the *I* filters, this *OFF* function reaches a higher peak after high intermittency signals.

The novelty function *N(t)* was defined as

$$N(t) = A_t e^{-\frac{t-t_L}{\tau_{Nd}}} \tag{11}$$

where $t_L$ is the time of the latest pulse onset. For the time between the onsets of the first and second pulse, $A_t := 1$. Otherwise, $A_t = 1 - e^{-(t_L - t'_L)/\tau_N}$, where again $t_L$ is the time of the latest pulse onset and $t'_L$ is the onset time of the pulse before the latest pulse. For a square wave signal, $t_L - t'_L$ becomes $T = \frac{1}{f}$, i.e. the period of the signal. Thus at first pulse onset, *N* spikes to 1 and decays with timescale $\tau_{Nd}$. At subsequent pulses, N spikes to a height of $A_t$ before decaying with timescale $\tau_{Nd}$. The $\tau_N$ timescale defines the time required between pulses to induce a strong response—if $T \ll \tau_N$, $A_t \approx 0$ and the novelty response is suppressed. On the other hand if $T \gg \tau_N$, $A_t \approx 1$ and the novelty response is maximal.

## Parameter Estimation

To estimate parameters for turn rate (Eq 2), we considered time points where flies were in fixations (i.e. not turning) or had just transitioned from fixation to turn. We excluded fixations that lasted longer than 1.5s, as more than 95% of fixations were shorter than 1.5s and beyond this duration, fixation durations were no longer exponentially distributed (**S8 Fig**). For a time point *t*, the probability to not initiate a turn at that time point is $e^{-\lambda_t \Delta t}$ where $\Delta t$ is our sampling time (1/60s) and $\lambda_t$ denotes the turn rate from Eq 2 at that time point. The probability to initiate a turn at that time point is $1 - e^{-\lambda_t \Delta t}$. We then constructed a likelihood function:

$$L = \Pi_{\text{fixations}} e^{-\lambda_t \cdot \Delta t} \Pi_{\text{turn starts}} (1 - e^{-\lambda_t \cdot \Delta t}) \tag{12}$$

and minimized the negative log of this likelihood using scipy.optimize.minimize with the standard L-BFGS-B method for minimization with bounds. All subsequent log-likelihood functions were minimized similarly. Note that for all timescales we estimated the log of the inverse timescale, i.e. the log of the 'rate' and then converted that back into a timescale.

To estimate turn speed parameters, we calculated the mean angular speed for each turn and subtracted our minimum turn speed of 25 deg/s (this is added back later when simulating the turns). The distribution of the resultant mean angular speeds was assumed to be a Gamma distribution with fixed shape parameter 2, based off the observations in **Fig 3**. The mean of this Gamma distribution was assumed to depend on the signal and is given by Eq 3. Assuming individual turns are independent, we could then construct a likelihood function as the product of the likelihood of each mean turn speed:

$$L = \Pi_{\text{turns}} \frac{1}{\left(\frac{\mu_t}{2}\right)^2} x_t \cdot e^{-\frac{x_t}{\frac{\mu_t}{2}}} \tag{13}$$

where $x_t$ is the observed mean angular speed (after subtracting 25 deg/s) for the turn and $\mu_t$ is the predicted mean angular speed for a given set of parameters, using Eq 3. As the timescales for the *N* and *OFF* responses were already estimated from the turn rate analysis, only the three $\mu$ coefficients were estimated from this likelihood function, by minimizing its negative logarithm.

For turn duration, for each turn we calculated its duration and subtracted the minimum turn duration, 0.18s (this is added back later when simulating turns). We then assumed an exponential distribution for these resultant turn durations and computed a likelihood function

as

$$L = \Pi_{\text{turns}} \lambda_{dur} e^{(-x_t \lambda_{dur})} \tag{14}$$

where here $x_t$ denotes the turn duration (after subtracting 0.18s). We found the constant $\lambda_{dur}$ that minimized the negative log-likelihood. The inverse of this constant is reported as $\tau_{dur}$ in Materials and Methods and Table 1.

For the turn bias (Eq 4), we first fit the $a_0$ parameter as explained in the main text. Given that the elevated upwind bias returns to baseline within a couple of seconds of ON block offset (**Fig 5C**, grey and **S12**, grey), we assumed that the bias in the last 10 s of the OFF block (**Fig 5D**, black) had no remaining signal dependence and so takes the form $1/(1+\exp[-a_0 \cdot \sin^2 \theta])$. We then minimized the squared error between this functional form and the no-signal turn bias curve obtained from data (**Fig 5D**, black), using scipy's optimize.least_squares routine and its default method, the Trust Region Reflective algorithm. The remaining turn bias parameters (timescales and gain factors in Eqs 4–9) were fit by simulating 240 flies (roughly how many trajectories were in the experiment) executing our full turning strategy (see below for simulation details) with all other parameters fixed to their fit value and minimizing the squared error between the observed mean $\theta(t)$ and predicted $\theta(t)$ over the first 20s of the experiment. The minimization was done with a brute-force search over the parameter space, where $g$ was discretized to 20 values linearly spaced between bounds shown in Materials and Methods and Table 1, while for the timescales we fit the log of the rates (i.e. $1/\tau$) by considering 20 values linearly spaced between the log of the minimum rate and log of the maximum rate, corresponding to fitting the timescales with logarithmic spacing. The parameters that minimized the mean squared error were used.

## Analysis of two-timescale integrating model

Following our analysis of a single timescale integrator $I(t)$ in [31], we consider the response $R(t)$ to a binary square-wave signal with frequency $f$ and duration $D$. If we let $R_n$ denote the value of $R(t)$ at the onset of the $n$th pulse, then by straightforwardly integrating Eqs 8 and 9, we get the relation

$$R_{n+1} = \left( R_n \cdot e^{-\frac{D}{\tau_g}} + 1 - e^{-\frac{D}{\tau_g}} \right) \cdot e^{-\frac{\frac{1}{f} - D}{\tau_d}} \tag{15}$$

Expanding and simplifying, we get

$$R_{n+1} = k \cdot R_n + k \cdot \left( e^{\frac{D}{\tau_g}} - 1 \right) \tag{16}$$

where we define $k = e^{-\frac{D}{\tau_g}} \cdot e^{-\frac{1}{f\tau_d}} \cdot e^{\frac{D}{\tau_d}}$. We can then see that in general

$$R_n = R_0 k^n + k \left( e^{\frac{D}{\tau_g}} - 1 \right) \cdot \frac{1 - k^n}{1 - k} \tag{17}$$

Note that $\frac{1}{f} = T$, the period of the square wave and so $k = e^{-\frac{D}{\tau_g}} \cdot e^{\frac{D-T}{\tau_d}}$, which we can see is less than 1. If we denote the asymptotic value of $R_n$ as $R_n^*$, we get

$$R_n^* = \left( e^{\frac{D}{\tau_g}} - 1 \right) \cdot \frac{k}{1 - k} \tag{18}$$

We can then compute the asymptotic average value of $R(t)$ over one period of the signal, which we denote as $\bar{R}$ as

$$\bar{R} = \frac{1}{T} \cdot \left[ \int_0^D R_n^* \cdot e^{-\frac{t}{\tau_g}} + 1 - e^{-\frac{t}{\tau_g}} \, dt + \int_0^{\frac{1}{f}-D} (R_n^* \cdot e^{-\frac{D}{\tau_g}} + 1 - e^{-\frac{D}{\tau_g}}) \cdot e^{-\frac{t}{\tau_d}} \, dt \right] \tag{19}$$

If we note that $f \cdot D = Int$, the intermittency of the signal, we get

$$\bar{R} = \frac{e^{\frac{Int}{f\tau_g}} \cdot \left( Int \cdot \left( e^{\frac{Int}{f\tau_d}} - e^{\frac{1}{f\tau_d}+\frac{Int}{f\tau_g}} \right) + f \cdot \left( e^{\frac{Int}{f\tau_d}} - e^{\frac{1}{f\tau_d}} \right) \cdot \left( e^{\frac{Int}{f\tau_g}} - 1 \right) \cdot (\tau_d - \tau_g) \right)}{\left( e^{Int\left(\frac{1}{\tau_d}+\frac{1}{\tau_g}\right)} - e^{-\frac{1}{f\tau_d}+\frac{2Int}{f\tau_g}} \right)} \tag{20}$$

We can see here that the response depends independently on both the intermittency and frequency of the signal. Since the response is integrating the signal (Eqs 8 and 9), $\bar{R}$ increases with signal intermittency. To see how it depends on frequency, we consider the difference between the timescales. Firstly, note that if $\tau_g = \tau_d$ then we just have a single timescale integrator and $\bar{R} = Int$, as we would expect from [31]. For the case where $\tau_g \ll \tau_d$ and can be approximated as 0 (as in our fits), one can readily compute $\bar{R}$, noting that $R(t)$ is 1 when the signal is present and decays with timescale $\tau_d$ when the signal is absent. We get

$$\bar{R} = Int + f\tau_d \left( 1 - e^{\frac{Int-1}{f\tau_d}} \right) \tag{21}$$

which we can see grows with $f$ until $f \gg 1/\tau_d$ at which point it levels off.

On the other hand, if $\tau_d \ll \tau_g$ and is taken to be 0 instead, we get

$$\bar{R} = Int + f\tau_g \left( e^{\frac{-Int}{f\tau_g}} - 1 \right) \tag{22}$$

and in this case decreases with increasing frequency, before leveling off. Thus we see that a short rise timescale and longer decay timescale are necessary for a positive response to both intermittency and frequency.

## Simulating fly turning dynamics

To simulate flies executing our turn model, we first determined whether a simulated fly would initiate a turn or not by taking the probability to initiate a turn in a time step $\Delta t$ as $\lambda(t) \cdot \Delta t$, where $\lambda(t)$ is given by Eq 2. If a fly was not turning its angular velocity was assumed to be 0. If a turn was initiated, its duration in excess of 0.18s was sampled from an exponential distribution with timescale $\tau_{dur}$, and added to 0.18s to get the total turn duration. The mean angular speed of the turn in excess 25 deg/s was sampled from a Gamma distribution with shape parameter 2 and mean given by Eq 3, then added to 25 deg/s to get the total mean angular speed. We assumed a turn had a parabolic angular speed profile (see Fig 3B). Given the mean value and duration (i.e. time between two zeros) of this parabola, we could compute $|\dot{\theta}|$ for the duration of the turn as $\left| \frac{6\mu}{d^2} (t - t_{start})(t - (t_{start} + d)) \right|$ where $t_{start}$ is the start time of the turn, $\mu$ is the mean angular speed of the turn, $d$ is the duration of the turn and the factor in front ensures that the average angular speed over the turn is equal to $\mu$. To determine the sign of the turn, we determined whether it was upwind or downwind by simulating a Bernoulli variable with probability given by Eq 4. Once $\dot{\theta}$ was specified for the whole turn, we could use Euler integration with timestep $\Delta t = 1/60$s (the frame rate of our experiments) to evolve a simulated agent's heading. Values for response functions $u(t)$ except for $F(t)$ were computed by Euler

integration with a step-size of $1/10^{th}$ our sampling rate and then resampled for agent simulation. $F(t)$ was computed as in [31]. Agents were initialized with headings sampled from the distribution of experimental flies' initial headings for that environment and initialized to all not be turning for simplicity (thus the first 0.1s of our simulated total angular speed is not shown in Figs 5C and 6 as the comparison with data would not be fair).

## Agent-Based Simulations in Complex Environments

To simulate odor plumes, we followed the procedure described by [44] and used by us in [31] where odor plumes are simulated as Gaussian concentration packets that are released at a Poisson rate, advected by a mean downwind velocity $v_d$, grow due to the effects of molecular diffusivity and small eddies and are typically perturbed in the crosswind direction by an eddy diffusivity. In our case, to account for correlated motion [45], we treated crosswind perturbations as a telegraph process, where packets' crosswind velocities would switch between $+v_c$ and $-v_c$ at a Poisson rate $\lambda_c$. Thus the period of time that a packet would sustain its crosswind velocity before switching was distributed exponentially with parameter $\lambda_c$. We used $v_d$ as 90mm/s, $v_c$ as 30mm/s and $\lambda_c$ as 2/s. The packet release rate for the low frequency plume was 0.75/s.

To generate plumes of increasing frequency in Fig 8E, all that was varied was the release rate. The following release rates were used: 0.75/s, 1.0/s 1.25/s, 1.5/s, 1.75/s, 2.0/s, 3.0/s, 5.0/s, 7.0/s. All other parameters were set as in [31] so that a concentration value of 1(a.u) was a reasonable detection threshold. To compute the mean plume frequency in these environments, we computed odor concentration (details below) at 10,000 points uniformly distributed in a circular sector with apex at the odor source and half angle $\arctan\left(\frac{1}{3}\right)$, set by the ratio of $v_c/v_d$. The radius of the sector was 250mm, which was the maximum downwind distance from the source where agents could be initialized.

To compute odor concentration values in these simulated environments, we assumed agents sensed odor in an elliptical region with semi-major axis of 0.75mm and semi-minor axis of 0.25mm, as in [27] and [31], with the major axis perpendicular to the agent's heading direction. Within this region individual points to sample odor concentration were chosen with a spacing of 6.5 points per mm, as this would match the spatial resolution of video smoke plumes that we have used in the past. This region was then evenly divided into a left region and a right region and odor signal for each region was computed by averaging the contribution from all packets, (where each packet's concentration decayed as a Gaussian with distance from the packet centroid) over the points in that region. We denote the left concentration as $C_L$ and the right concentration as $C_R$. The odor signal was then computed as $\frac{C_L + C_R}{2}$.

The odor motion signal was computed as $C_L(t - \Delta t) \cdot C_R(t) - C_L(t) \cdot C_R(t - \Delta t)$, in accordance with a basic Hassenstein-Reichardt correlator and as used in [28]. $\Delta t$ was taken to be one time step of the simulation. A motion signal threshold was set at 0.01. If the signal was above 0.01, it meant that motion had been detected from the left, and so the against-motion direction would be the agent's current heading plus 90 degrees. Similarly, if the signal was below -0.01 it meant motion from the right and so the against-motion direction was the agent's heading minus 90 degrees. From this the bias direction was computed as shown in in **Fig 8A**.

For all simulations, agents' initial positions were uniformly distributed in a region from x = 200mm to 250 mm and y = -60mm to 60mm, where x denotes the downwind direction and y the crosswind direction (see **Fig 8C**). Agents' orientations were uniformly initialized between 90 degrees and 270 degrees (where 180 degrees was the upwind direction). The source position was x = 10mm and y = 0mm. The simulation time was set to 75s with time step of 1/

60s. The success region as denoted in **Fig 8C** was a square region from x = 0 to x = 25mm and y = -12.5mm to y = 12.5mm.

To compute error bars in **Fig 8D** and **8E**, once a binary success vector was obtained from simulation (so a vector of size 100,000 that was 1 if the agent at that index reached the source region in the simulation time and 0 otherwise), it was resampled with replacement to generate 1000 other such vectors. The standard deviation in the mean of these vectors is reported as the error. To compute the error of the ratios in **Fig 8E**, a similar procedure was used except that two sets of 1000 resampled vectors were generated (one set for agents with novelty modulation, the other for agents without) and from this, 1000 estimates of the ratio of success rates were computed. The standard deviation of these 1000 ratios was reported as the error. Raw success rates ranged from around 3.5% in the low frequency plume to 25% in the highest frequency plume. The success rate in the no odor control was around 0.5%.

### Estimating stopping and walking rates

A stop was defined as a period of time where a fly's walking speed was less than 2mm/s for more than 100ms. A walk was when walking speed was greater than 2mm/s for more than 100ms. The rates for transitioning from stopping to walking and walking to stopping were then estimated and smoothed in the same manner as the turn rate, where the fraction of flies transitioning in a single timestep $\Delta t$ was taken to be $\lambda \Delta t$, where $\lambda$ denotes the relevant rate. The uncertainty is computed by bootstrapping in the same way as for the turn rate estimation.

Fly lines used in experiments are available upon request.

## Supporting information

**S1 Fig. In absence of fictive or real odor, flies orient on average crosswind in both laminar and complex wind. A.** Population mean orientation of flies in absence of odor stimulus, in laminar (green) or complex (purple) wind. Wind was presented continuously for 5 minutes. *Orco>Chr*: *w; gmr-hid/+; Orco-GAL4/UAS-20XChrimson* (experimental line responsive to fictive odor stimulus). GAL4 PARENT: *w;+;Orco-Gal4*. UAS PARENT: *w;gmr-hid;UAS-20XChrimson*. Canton-S: wildtype (same as in [27]). Recorded at 60 fps, but presented by sub-sampling every 5 frames. Dashed line indicates orienting crosswind (90˚). All flies that moved less than 2mm/s on average for their entire trajectory were removed. Orco>Chr: laminar, n = 5–69 trajectories per time point across the entire 5 minute recording; complex, n = 8–56. GAL4 PARENT: laminar, n = 1–18; complex, n = 1–17. UAS PARENT: laminar, n = 1–13; complex, n = 1–12. Canton-S: laminar, n = 1–13; complex, n = 1–12. **B.** Population orientation mean and SEM across all trajectories for laminar (green) and complex (purple) wind without odor stimulus over 5 minute recording. Any time points where the fly moved less than 2 mm/s were removed. Orco>Chr: laminar, 91.1˚ ± 1.1˚, n = 678 trajectories; complex, 91.9˚ ± 1.4˚, n = 519. GAL4 parent: laminar, 84.1˚ ± 2.9˚, n = 152, complex, 82.2˚ ± 3.3˚, n = 119. UAS parent: laminar, 87.2˚ ± 3.7˚, n = 102, complex, 89.8˚ ± 3.0˚, n = 114. Canton-S: laminar, 90.2˚ ± 4.4˚, n = 120, complex, 90.3˚ ± 4.3˚, n = 111.
(EPS)

**S2 Fig. Parent flies and Orco>Chr flies without ATR do not respond to the fictive odor signal.** Traces of mean orientation over time for control lines in presence of 2 Hz, 0.1 s fictive odor. Population mean orientation of GAL4 parent (left) and UAS parent (middle) genotypes of optogenetically active flies, and the optogenetically active line without feeding ATR (right). Black line is mean across all trajectories at that time point, grey shading indicates SEM. Dashed black line indicates crosswind orientation at 90˚. Red bars indicate fictive odor presence. Any

time points where the fly moved less than 2 mm/s were removed. GAL4 parent: 52–74 trajectories per frame, UAS parent: 50–89 trajectories per frame, Orco>Chr (no ATR): 108–159 trajectories per frame.
(EPS)

**S3 Fig. Population mean behavioral responses of flies navigating fictive odor stimuli are similar in laminar and complex wind.** Mean population orientation response measured from flies navigating one of 6 from the 45 fictive odor environments: 0.2 Hz 1 s, 0.5 Hz 0.1s, 0.5 Hz 1s, 1 Hz 0.5 s, 2 Hz 0.1 s, 2 Hz 0.25 s (from left to right), in either laminar (green) or complex (purple) wind structure. 235–485 trajectories were recorded per odor and wind environment, and 104–242 trajectories were recorded per time frame (recording rate = 60 frames per second). Red bars indicate odor presence.
(EPS)

**S4 Fig. Instantaneous angular velocity as a function of environment duration and intermittency. A.** Instantaneous angular velocity of flies as a function of their orientation during the ON block (0–15 s). Upwind orientation is at 180°, downwind at 0°. Orientation was split into 8 bins with width 22.5°. Vertical dashed line at 90° indicates flies facing crosswind during ON block. Positive angular velocity represents turning upwind, negative represents turning downwind. Horizontal solid line at 0°/s indicates no change in angular velocity, thus no change in orientation. Angular velocity is colored by environment duration. Blue: 0.02 s, indigo: 0.05 s, magenta: 0.1 s, red: 0.25 s, orange: 0.5 s, yellow: 1.0 s. The grey bar indicates the crosswind range (90° ± 22.5°) over which the mean angular velocity during the ON block per odor environment was calculated in **Fig 2C**. **B.** Same as **A** but colored by environment intermittency. Blue: 0–0.05, purple: 0.05–0.2, red: 0.2–0.5, yellow: 0.5–0.875.
(EPS)

**S5 Fig. Angular speed responses vary across stimulus frequency and intermittency.** Population mean instantaneous angular speed, obtained from the absolute of the angular velocity, across 45 fictive odor environments, derived from the orientation (see Materials and Methods). Stimuli presented are the same as in **Fig 2A**. Red bars denote the signal simultaneously encountered by all flies within an experiment. Between 176 and 407 trajectories were recorded per environment. Between 72 and 237 trajectories were recorded per time point across all environments at 60 fps.
(EPS)

**S6 Fig. Angular speed response is consistent across fly orientations: Angular speed from 4 of the 45 odor environments (0.2 Hz 1 s, 0.5 Hz 0.25 s s, 1.5 Hz 0.1s s, 3 Hz 0.25 s).** Heading was split into three bins of 60 degrees; upwind facing (120°-180°, blue), crosswind facing (60°-120°, orange) and downwind facing (0°-60°, green). Orientation was flipped over 180° as before. Solid line: population mean angular speed of flies oriented within the corresponding 60° bin at each time point. Lighter shading: mean ± SEM at each time point. Red bars: fictive odor pulses. Between 170 and 305 trajectories were recorded per environment for each 60° heading bin. Between 9 and 105 trajectories contribute to the data at each time point.
(TIFF)

**S7 Fig. Defining turn events. A.** The 95th percentile of the distribution of angular change magnitude during fixation events, $\Delta\theta_{95}$, as a function of the angular speed threshold set to define a turn event. Thresholds tested ranged from 5 deg/s to 150 deg/s. Red dashed line highlights the inflection point in the curve at ~25 deg/s, indicating that thresholds greater than 25 deg/s ignore larger angular changes that could contribute to changes in heading. **B.** After

applying the 25 deg/s threshold angular speed to turn events, the distribution of angular change during turns is bimodal: there is a shorter range distribution centralized around small mean angular changes, and a much wider distribution of angular changes around centralized around a larger mean angular change. The smaller changes are less likely to influence heading compared to the larger angular changes, thus we sought to exclude them. Distribution can be fit using a Gaussian mixture model with a low mean, low variance Gaussian and a high mean, high variance Gaussian. The standard deviation of the low-mean Gaussian is 4.5˚, which we used to set a minimum turn event duration of 0.18 s, given a minimum turn angular speed of 25 deg/s. **C.** After applying the 0.18 s threshold duration for a turn event, the distribution of angular change is no longer bimodal, meaning that only turn events of significant magnitude are selected.
(EPS)

**S8 Fig. Defining turn rate. A.** Distribution of fixation durations (bin number = 60), having applied the turn speed and turn duration thresholds. Slope of the distribution (i.e. the turn rate) is non-linear; fixation durations can reach up to ~40 s. **B.** Cumulative probability density function of fixation durations. Red line indicates 95th percentile of fixation durations. Vertical line indicates that more than 95% of fixation events have a duration of less than 1.5 s. **C.** Distribution of fixation durations, for fixations lasting up to 1.5 s. These are exponentially distributed, allowing us to model turn rate as a Poisson process.
(EPS)

**S9 Fig. Turn rate is dynamic and well-predicted by the combination of novelty and offset responses.** Experimental population turn rate estimated with a sliding 0.25s window (grey line) and bootstrapped error (grey shading) (see Materials and Methods) for all 45 fictive odor environments as well as model predictions (pink) and simulated error (pink shading) using Eq 2 and fit with one fixed set of parameters to all 45 environments using Maximum Likelihood Estimation (see Materials and Methods). Turn rate is dynamic and reasonably well-approximated by the model. Each panel had ~150 trajectories contributing at any timepoint. To generate model predictions and error, for each panel we generated ~150 turn series (1 if a turn starts at that time point, 0 if not) and averaged and smoothed this data to get a single time-varying turn rate estimate. We repeated this process 10000 times, taking the mean of all these as the model turn rate and the standard deviation as the model error. For more details see Materials and Methods. N = 8–103 turn events per time point for all 45 environments.
(EPS)

**S10 Fig. Mean turn speed is also dynamic and well-predicted by the combination of novelty and offset responses.** Experimental population mean angular speed given turning estimated with a sliding 0.25s window (grey line) and bootstrapped error (grey shading) (see Materials and Methods) for all 45 fictive odor environments as well as model predictions (pink) and simulated error (pink shading) using Eq 3 and fit with one fixed set of parameters to all 45 environments using Maximum Likelihood Estimation (see Materials and Methods), and using timescales for *N* and *OFF* responses extracted from the turn rate parameter estimates. Angular speed given turning is dynamic and reasonably well-approximated by the model. To generate model predictions and error, artificial turns were simulated occurring at the same times as the real turns. The speed of these turns in excess of the 25 deg/s threshold was sampled from a gamma distribution with time varying mean, given by Eq 3. Average turn speed on this simulated data was then calculated with the same sliding 0.25s window. We repeated this process 10000 times, taking the mean of all these as the model turn angular speed and the standard deviation as the model error. For more details see Materials and Methods. Ns are the same as in S9 Fig.
(EPS)

**S11 Fig. Mean turn duration is roughly constant in time for high frequency environments but shows a small signal dependency for low frequency long duration signal:** Mean turn duration (grey) and estimated error (grey shading) vs. time for all 45 environments considered as well as the MLE estimate of a constant duration (pink line) and uncertainty (pink shading) (see Materials and Methods). We do see some signal modulation of turn duration, particularly at lower frequencies and often see an increase in turn duration at the offset of the ON block. However, the strength of this modulation is rather mild overall and the response at block offset is somewhat unpredictable and not well fit with the off response timescales found for turn rate. Thus, to keep our model as simple as possible we neglected modulations in turn duration. To generate model predictions and error, artificial turns were simulated occurring at the same times as the real turns. The duration of these turns in excess of the 0.18s threshold was sampled from an exponential distribution with fixed parameter. Average turn speed on this simulated data was then calculated with the same sliding 0.25s window. We repeated this process 10000 times, taking the mean of all these as the model turn duration and the standard deviation as the model error. For more details see Materials and Methods. Ns are the same as in S9 Fig.
(EPS)

**S12 Fig. Turn bias is dynamic and well approximated by a two-timescale integrator with an instantaneous rise timescale.** Probability to turn upwind given turning (i.e. upwind bias) vs time (grey) and estimated error (grey shading) for all 45 fictive odor environments along with model predictions (purple) and estimated error (purple shading) for a two-timescale integrating response (Eqs 8–9) with instantaneous rise timescale and finite decay timescale (see Materials and Methods). Upwind bias was estimated by calculating the fraction of turns that were upwind in a sliding 0.25s window. We see that upwind bias generally rises during signal presence and otherwise is slightly less than 0.5, suggesting that in the absence of signal flies display a preference for downwind orientation. To generate model predictions and error, artificial turns were simulated occurring at the same times as the real turns, with the same initial orientations. The direction of these turns (upwind or downwind) was assigned according to the probability given by the time varying bias, Eq 4. Turn bias on this simulated data was then calculated with the same sliding 0.25s window. We repeated this process 1000 times, taking the mean of all these as the model turn bias and the standard deviation as the model error. For more details see Materials and Methods. N = 8–103 turn events per time point for all 45 environments.
(EPS)

**S13 Fig. Walking speed and stopping behavior in temporally diverse fictive odor environments: A)** Average walking speed of tracked agents in the different temporal environments. Here we exclude flies with speeds less than 2mm/s since that is the threshold for being considered as stopped. Grey shading denotes SEM. Between 47 and 205 trajectories contribute to each time point. **B)** Stop-to-walk transition rate (blue) and walk-to-stop transition rate (orange) as a function of time (Materials and Methods for more details on estimating these rates). Between 8 and 86 trajectories contribute to the stop-to-walk rate at each time point and 37 and 208 contribute to the walk-to-stop rate, before smoothing. For **A)** and **B)** the red bars denote the fictive odor signal as in other figures.
(EPS)

## Acknowledgments

We thank J. Jeanne, D. A. Clark, G. M. Santana, M. Demir, H. Mattingly, and K. Kamino for helpful discussions.

## Author Contributions

**Conceptualization:** Viraaj Jayaram, Aarti Sehdev, Nirag Kadakia, Thierry Emonet.

**Data curation:** Viraaj Jayaram, Aarti Sehdev, Ethan A. Brown.

**Formal analysis:** Viraaj Jayaram, Aarti Sehdev, Nirag Kadakia, Thierry Emonet.

**Funding acquisition:** Thierry Emonet.

**Investigation:** Viraaj Jayaram, Aarti Sehdev, Ethan A. Brown.

**Methodology:** Viraaj Jayaram, Aarti Sehdev, Nirag Kadakia, Thierry Emonet.

**Project administration:** Thierry Emonet.

**Resources:** Thierry Emonet.

**Software:** Viraaj Jayaram, Aarti Sehdev, Nirag Kadakia.

**Supervision:** Thierry Emonet.

**Validation:** Viraaj Jayaram, Aarti Sehdev, Nirag Kadakia, Ethan A. Brown, Thierry Emonet.

**Visualization:** Viraaj Jayaram, Aarti Sehdev, Nirag Kadakia, Thierry Emonet.

**Writing – original draft:** Viraaj Jayaram, Aarti Sehdev, Nirag Kadakia, Thierry Emonet.

**Writing – review & editing:** Viraaj Jayaram, Aarti Sehdev, Nirag Kadakia, Thierry Emonet.

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
