## [Decision Letter · Decision Letter 0]

5 Dec 2022

Dear Dr. Emonet,

Thank you very much for submitting your manuscript "Novelty detection and multiple timescale integration drive Drosophila orientation dynamics in temporally diverse olfactory environments" for consideration at PLOS Computational Biology. As with all papers reviewed by the journal, your manuscript was reviewed by members of the editorial board and by several independent reviewers. The reviewers appreciated the attention to an important topic. Based on the reviews, we are likely to accept this manuscript for publication, providing that you modify the manuscript according to the review recommendations.

Specifically, the reviewers appreciated the experimental and analysis approach to isolate the effects of temporal factors in driving Drosophila olfactory navigation but felt that the modelling component would be strengthened by consideration of how the temporal effects highlighted here interact with other known factors (e.g. spatial, concentration) to produce actual odour trajectories. Note it is not expected in this revision that the direct effect of simultaneously altering intensity and timing be examined in new experiments (as reviewer 2 describes) but, for example, modelling be used to make predictions of whether and how this might be expected to influence behaviour.

Sincerely,

Barbara Webb

Academic Editor

PLOS Computational Biology

Marieke van Vugt

Section Editor

PLOS Computational Biology

Reviewer's Responses to Questions

**Comments to the Authors:**

Reviewer #1: This manuscript by Sehdev…Emonet and colleagues builds on recent work from their group and others to further quantify the temporal cues in odor that drive olfactory navigation behaviors in Drosophila melanogaster. They use optogenetic activation of olfactory receptor neurons to examine behavioral responses to pulse odor stimuli of different frequencies and durations, and fit models to both the upwind turning component and a non-directed turn component evoked by onset and offset of the optogenetic stimulus. The main new findings are (1) a novelty non-directed turning response to a stimulus presented after a significant break, and (2) a refinement of a model from their previous paper (Jayaram et al) of the upwind turning response, arguing that a two-timescale integrator model fits the data somewhat better that the their previous model that linearly combines intermittency and frequency sensing. Overall, while I think the experiments and modeling are carefully done, I am concerned that the advance on their previous work is somewhat limited. In particular, the models presented in this paper do not incorporate the contribution of odor motion sensing that this group recently described, nor any possible contribution of spatial sensing to olfactory navigation (e.g. from Bhandawat lab work). Because they show model fits only to mean parameters of the behavioral data (e.g. turn rate or turn speed) and not simulations of full trajectories, it is not clear how significant the differences in the model from their previous models are for the ultimate navigation success of the animal, not whether the difference between the four models they propose for the temporal integration underlying upwind turning significantly exceeds the differences between individuals, which can be substantial. Although the scope of the paper is up to the authors, I feel that it would be more impactful for the field to put forward models that incorporate spatial and spatio-temporal contributions of odor sensing to navigation, in addition to temporal ones, and to make provision for understanding the role of individual differences in olfactory navigation behavior.

Major concerns:

1) The authors demonstrate the validity of their model with the ability recapitulate mean behavioral parameters (e.g. Figure 5D Fig. 6), but not he ability to capture individual variability or the trajectories of real flies. I think it would be useful to use their models to simulate real trajectories and evaluate whether the differences between this model and their/others previous models significantly impacts navigation success in different olfactory environments. In addition, it would be helpful to know which parameters of the model(s) account for differences across individuals. Several figures illustrating variation across individuals are unclear (see note on Fig. 1C below).

2) The authors make several references to their exciting recent work showing the contribution of odor motion to navigation but odor motion effects on navigation do not form a part of the models presented here. How does odor motion sensing combine with the temporal sensing described here and what are the relative weights of these different factors? It seems like they should be able to address this using existing data from this and the previous paper.

3) The use of optogenetic activation to produce navigation behaviors in flies is quite well established at this point (e.g. Bell…Wilson, Schulze..Louis, Tao…Bhandawat, Tumkaya…Clardig-Chang, Fox…Nagel, Matheson…Nagel), so I think the material in Figure 1 could be gone through more quickly.

4) The Discussion focusses almost exclusively on the adult antennal lobe, and does not mention significant experiments and models in the larval antennal lobe (e.g. Schulze…Louis, Gepner…Gershow) which likely performs similar temporal processing, nor work on central circuits for navigation in both larvae and adults (e.g. Tastekin…Louis, Matheson…Nagel) which could present additional loci for the temporal computations they identify here.

additional:

Fig. 1C: I am confused by this plot which I think shows mean orientation. Why are the errorbars not larger during the baseline (pre-odor) phase compared to odor? This figure seems to imply that all flies are fairly well oriented around 90° in the absence of stimulus, to the same degree that they are oriented at 135° during odor, which seems implausible to me. Are the flies shown here selected for a particular pre-odor orientation? Otherwise perhaps a histogram of orientations during the different phases would help to show variation in behavior across individuals.

line 145: gmr-hid. Can you please spell out what this is and what it does?

line 159-165. This paragraph should note the many other studies that have used opto-genetic stimulation to evoke navigation behaviors (Tao/Bhandawat, Tumkaya/Gepner, Matheson/Nagel).

line 266: angular speed dynamics depended more on changes in turn rate and turn speed…”. This would seem to conflict with the statement in Demir et al that navigation does not depend on changes in turn rate. please comment.

upwind turning model in Fig. 5B/C. For similarity with others literature, it might be helpful to plot this with the upwind point (0/180°) at the middle of the plot, illustrating that it forms a stable fixed point of the system. This would connect with literature on fixation in the visual system (e.g. Reichardt and Poggio, 1976) and recent studies in the central complex (e.g. Hulse et al. 2020, Goulard and Webb, 2021, Matheson/Nagel 2022).

line 585: “for a simple integrating response to increase independently with frequency and intermittency, it is necessary for the rise timescale to be faster than the decay timescale” — doesn’t this have to be true?

line 604: “we did not factor changes in ground speed or transitions between stops and walk bouts”. These have both been modeled before by this group and others—-why not incorporate these into the model?

Reviewer #2: This is an elegant study, and I am very enthusiastic about the manuscript and its results. I only have one minor comment/question for the authors. In many moth species, the flux of the odor molecules has been shown to be critical for odor-mediated navigation. The optogenetic activation of the orco pathway in Drosophila allows precise control of the temporal dynamics of odor "encounter", but I'm wondering in this study if the authors have examined the interplay between concentration~encounter. The authors' eLife article touches on this aspect, but I'm wondering if it's possible to also examine this by scaling the temporal dynamics of the photostimulation with the intensity of the light stimulation as a proxy for odor concentration.

**Have the authors made all data and (if applicable) computational code underlying the findings in their manuscript fully available?**

Reviewer #1: **No: **manuscript says they are "available upon request"

Reviewer #2: **No: **The authors should place their code and data on github (immediately after the manuscript is accepted)

PLOS authors have the option to publish the peer review history of their article (what does this mean?). If published, this will include your full peer review and any attached files.

Reviewer #1: No

Reviewer #2: No

Figure Files:

Data Requirements:

Reproducibility:

References:

---

## [Decision Letter · Decision Letter 1]

20 Mar 2023

Dear Dr. Emonet,

Thank you very much for submitting your manuscript "Temporal novelty detection and multiple timescale integration drive Drosophila orientation dynamics in temporally diverse olfactory environments" for consideration at PLOS Computational Biology. As with all papers reviewed by the journal, your manuscript was reviewed by members of the editorial board and by several independent reviewers. The reviewers appreciated the attention to an important topic. Based on the reviews, we are likely to accept this manuscript for publication, providing that you modify the manuscript according to the review recommendations.

Sincerely,

Barbara Webb

Academic Editor

PLOS Computational Biology

Marieke van Vugt

Section Editor

PLOS Computational Biology

Reviewer's Responses to Questions

**Comments to the Authors:**

Reviewer #1: The authors have addressed my concerns. The new modeling adds to the manuscript and helps flesh out a fuller picture of how different elements are combined in Drosophila olfactory navigation.

A few minor points to add to the Discussion:

— Although the authors use optogenetic activation of orco to dissect the temporal dynamics of “odor” that drive components of behavior, I think there is strong evidence that different odors are processed differently in time to drive behavior. This was recently demonstrated elegantly by Zocchi…Hong (Current Biology, 2022) who showed that flies will walk upwind in a continuous stream of apple cider vinegar, but require pulse CO2 to show an upwind behavioral response. This is consistent with earlier findings in mosquitoes that different dynamics of human scent and CO2 drive approach behavior. I think this point should be included in the Discussion or Intro.

— another recent preprint (Aso…Hige, bioRxiv, 2022) identifies additional central circuitry that plays a part in upwind orientation behavior.

minor things:

line 29: maybe “the fraction”?

line 59: “incredibly challenging task” I dunno. Ermentrout has done some work showing that a variety of relatively simple algorithms can be successful in certain plume structures. Maybe just “challenging”?

line 84: “straight odor ribbons” not all of the cited papers used ribbons of odor.

line 93: frequency of odor encounters. see my point above about the Zocchi paper. I think it is unlikely that there is a single set of dynamics that all insects use to process odor, rather the dynamics of odor that evoke behavior is likely odor and species specific.

line 200: “Combination that produced indeterminable encounters were excluded.” I am not sure what you mean by this.

line 487: “For the intermittency sensing model (…) predicted well the response. This sentence is not grammatical.

Reviewer #2: The authors have performed an excellent study, and I am excited to see it published.

**Have the authors made all data and (if applicable) computational code underlying the findings in their manuscript fully available?**

Reviewer #1: Yes

Reviewer #2: None

PLOS authors have the option to publish the peer review history of their article (what does this mean?). If published, this will include your full peer review and any attached files.

Reviewer #1: No

Reviewer #2: No

Figure Files:

Data Requirements:

Reproducibility:

References:

---

## [Editor Report · Decision Letter 2]

3 Apr 2023

Dear Dr. Emonet,

We are pleased to inform you that your manuscript 'Temporal novelty detection and multiple timescale integration drive Drosophila orientation dynamics in temporally diverse olfactory environments' has been provisionally accepted for publication in PLOS Computational Biology.

Best regards,

Barbara Webb

Academic Editor

PLOS Computational Biology

Marieke van Vugt

Section Editor

PLOS Computational Biology

---

## [Editor Report · Acceptance letter]

4 May 2023

PCOMPBIOL-D-22-01414R2 

Temporal novelty detection and multiple timescale integration drive *Drosophila* orientation dynamics in temporally diverse olfactory environments

Dear Dr Emonet,

I am pleased to inform you that your manuscript has been formally accepted for publication in PLOS Computational Biology. Your manuscript is now with our production department and you will be notified of the publication date in due course.

With kind regards,

Zsofia Freund
